# Local and transboundary contributions to $NO_y$ loadings across East Asia using CMAQ-ISAM and GEMS-informed emissions inventory during the winter-spring transition

Jincheol Park, Yunsoo Choi, Sagun Gopal Kayastha

Department of Earth and Atmospheric Sciences, University of Houston, Houston, TX, USA

*Correspondence to*: Yunsoo Choi (ychoi6@uh.edu)

**Abstract.** We investigated source contributions of nitrogen oxides ($NO_x$) emissions to reactive nitrogen species ($NO_y$) loadings across East Asia during the 2022 winter-spring transition. Using the Community Multiscale Air Quality model and
its Integrated Source Apportionment Method, we conducted air quality simulations, leveraging top-down estimates of $NO_x$ emissions informed by the Geostationary Environment Monitoring Spectrometer tropospheric nitrogen dioxide ($NO_2$) columns. After our GEMS-informed Bayesian inversion, the inventoried $NO_x$ emissions increased by 50% in Korea and 33% in China, which substantially reduced the model's prior underestimation of surface $NO_2$ concentrations from -32.75% to -13.01% in Korea and from -10.26% to -3.04% in China. We compared local and transboundary contributions of $NO_x$
emissions to $NO_y$ concentrations across East Asia. Local contributions showed a declining trend, from 32%-43% in January to 23%-30% by May, while transboundary contributions consistently increased from 16%-33% in January to 27%-37% by May. North China consistently contributed over 10% to East Asia's $NO_y$ loadings. East China and South Central China were significant contributors to each other's $NO_y$ budget by 9%-12%. South Central China transboundary contributions consistently outweighed local contributions by 5%, indicating vulnerability to pollution transport. Korea, initially the least
influential, contributed 1%-4% to transboundary $NO_y$ concentrations in January. This increased to 6%–7% by May, becoming comparable to other regions' contributions. These behaviors of $NO_y$ were driven by distinct synoptic settings, where strong wintertime northwesterly winds directed pollutants southeastward, while their weakening in spring led to more multidirectional transport patterns, allowing pollutants to spread more broadly across the regions.

## 1 Introduction

Nitrogen oxides ($NO_x$) emissions have long been a significant concern in East Asia due to their detrimental impact on air quality and public health, particularly in densely populated urban areas (Hoek et al., 2013; Newell et al., 2018). $NO_x$ is also a major precursor of secondary aerosols, contributing to the formation of fine particulate matter ($PM_{2.5}$), exacerbating airborne health risks. $NO_x$ has a relatively short atmospheric lifetime, typically ranging from a few hours (Beirle et al., 2011; Lin et al., 2012; Liu et al., 2016; Lange et al., 2022) to several days (Tang et al., 2023; Goldberg et al., 2024), depending on

meteorological conditions and chemical regimes in the region. Despite this, $NO_x$ plays a significant role in transboundary pollution. For instance, $NO_x$ emissions from China have been shown to substantially contribute to air quality in Korea during transboundary transport episodes, driven by specific meteorological conditions and chemical regimes in the region. During its transport across the Yellow Sea, $NO_x$ undergoes chemical reactions, forming secondary aerosols and contributing to elevated particulate matter concentrations in Korea (Nault et al., 2018; Eck et al., 2020; Jordan et al., 2020). Furthermore, $NO_x$ rapidly transforms into longer-lived reactive nitrogen species, collectively known as $NO_y$, which can be transported across long distances. $NO_y$ species, such as nitric acid ($HNO_3$), nitrous acid (HONO), and peroxyacetyl nitrate (PAN), play a significant role in redistributing nitrogen across extensive regions (Hertel et al., 2012). This extends the impact of $NO_x$ emissions beyond their sources, contributing to both local and transboundary air quality challenges.

The transport of air pollutants across Asia, particularly pronounced during the winter-spring transition, is largely determined by shifting synoptic systems that influence both the direction and extent of pollutant movement. In winter, the Siberian High dominates the region, maintaining cold, stable high-pressure systems that drive strong northerly and northwesterly winds (Hui, 2007; Kim et al., 2013; Wyrwoll et al., 2016; Dong et al., 2020). These winds typically result in southeastward and southward pollutant transports, carrying air pollutants from densely industrialized areas in northern China toward downwind regions such as other parts of China, Korea, Southeast Asia, and Japan (Ikeda et al., 2015; Chen et al., 2021; Wu, 2021; Zhao et al., 2021; Gu et al, 2024; Kang et al., 2024). However, the Siberian High's stable nature can also induce significant subsidence, which traps pollutants in the lower boundary layer, limiting vertical mixing and causing pollutants to accumulate locally near their sources (Zhang et al., 2007; Zhai et al., 2024). As winter progresses into spring, the Siberian High weakens, and while this leads to weaker northwesterly winds, the directionality of pollutant transport becomes more complex due to changes in synoptic patterns. Spring introduces slowly traveling high and low-pressure systems along with shifting wind directions, leading to multi-directional movements of air pollutants across Asia (Peterson et al., 2019). This seasonal transition, along with enhanced vertical mixing from warmer air temperatures and weaker subsidence, allows pollutants to disperse more readily within the boundary layer, facilitating their transport across extended distances (Ryu and Min, 2024). This complexity makes it difficult to fully understand how region-specific pollutant emissions contribute to air quality across Asia, complicating efforts to develop effective air quality management strategies.

To better understand the transboundary behaviors of air pollutants, chemical transport models (CTMs) have been widely used. CTMs, such as the Community Multiscale Air Quality (CMAQ) model (Byun & Schere, 2006), translate emissions inputs and underlying meteorology into three-dimensional representations of air pollutant loadings. This enables simulations of local and transboundary air pollution dynamics, providing insights into the origins and transport pathways of air pollutants. For example, Dong et al. (2020) conducted source apportionment of surface $PM_{2.5}$ concentrations in the Beijing-Tianjin-Hebei (BTH) region, northern China, using CMAQ and its Integrated Source Apportionment Method (CMAQ-ISAM). From 2014 to 2017, annual mean $PM_{2.5}$ concentrations decreased by 33%, with local emissions reductions accounting for 47%,

intra-regional transport contributing 25%, and transport from outside the region contributing 28%. The contribution of regional transport increased by up to 40% during spring and winter, driven by strong northwesterly winds. Yang et al. (2021) conducted source apportionment of ozone concentrations during a severe spring 2020 ozone peak event in the Sichuan Basin, southwestern China, using CMAQ-ISAM. Initially, northeasterly winds transported ozone precursors, including $NO_x$ and VOCs, from the northern boundary of the region, contributing over 50% of the ozone in the basin. As the synoptic pattern evolved, southeasterly winds trapped ozone and its precursors within the basin, leading to elevated ozone concentrations from local emission sources. Similarly, Xian et al. (2024a)'s source apportionment study in the Sichuan Basin during the warm growing season of 2022, using CMAQ-ISAM, demonstrated that persistent northeasterly winds transported ozone precursors from outside the basin, contributing nearly 40% of the region's ozone concentrations, while the rest was driven by local precursor emissions. Bae et al. (2020) investigated the influence of $NO_x$ and sulfur dioxide ($SO_2$) emissions from China on $PM_{2.5}$ concentrations in the Seoul Metropolitan Area (SMA) of Korea during the years 2012 to 2016, using CMAQ. Long-range transport of $NO_x$ and $SO_2$, which served as precursors for secondary aerosols such as nitrate and sulfate, significantly contributed to aerosol loadings in the SMA. Nitrate aerosols, in particular, comprised 50% of $PM_{2.5}$ during winter and 67% during spring, underscoring their significant share of $PM_{2.5}$ pollution in the SMA during these seasons. Similarly, Lee et al.'s study (2020) demonstrated that the long-range transport of $PM_{2.5}$ and its precursors, including $NO_x$, $SO_2$, and volatile organic compounds (VOCs), originating from China led to substantial increases in $PM_{2.5}$ concentrations in Korea by up to 50% during the 2016 KORUS-AQ campaign period. Tang et al. (2023) also assessed the contributions of local and transboundary emissions to $PM_{2.5}$ concentrations in Korea during the 2016 KORUS-AQ campaign. Under stagnant high-pressure conditions, local emissions were the dominant source, accounted for up to 49% of $PM_{2.5}$ concentrations in the SMA. However, during periods of strong westerly winds, pollutants transported from China significantly impacted air quality in the SMA, contributing as much as 71% of $PM_{2.5}$ concentrations. This was primarily driven by secondary inorganic aerosols, with nitrate and ammonium aerosols contributing up to 51% and 70% of the total $PM_{2.5}$ mass, respectively. Kashfi Yeganeh et al. (2024) quantified the contributions of transboundary $NO_x$ and VOC sources to ozone concentrations in Seoul, Korea, during a June 2019 ozone exceedance event. Ozone precursors were transported by northwesterly and westerly winds from China, accounting for 57.7% of the ozone concentrations in Seoul, while local emissions contributed 42.3%. Gu et al. (2024) assessed the health impacts of surface $PM_{2.5}$ and ozone concentrations across Southeast Asian countries in 2018 using CMAQ-ISAM. Local emissions contributed up to 87% and 60% of $PM_{2.5}$- and ozone-related premature mortalities, respectively, while transboundary air pollution accounted for up to 13% and 40%. Beyond the studies mentioned above, a number of CTM-driven source apportionment efforts have highlighted the substantial contributions of both local and transboundary pollutants to air quality across Asia (Kajino et al., 2013; Wang et al., 2015; Li et al., 2017a; Li et al., 2019; Shen et al., 2022; Xian et al., 2024b). While such earlier efforts have provided valuable insights into Asia's pollution dynamics influenced by prevailing winds and dominant emission sources, the extent of source contributions varied across the regions due to differences in the materials and methods employed in each study. A common issue highlighted in many of these studies is the uncertainty in emissions inventories, which can compromise the reliability of simulations. This often

stems from the use of incomplete or outdated emissions inventories, as pointed out in several previous studies (Carmichael et al., 2002; Pan et al., 2014; Sargent et al., 2021; Russo, 2019; Han et al., 2021; Liu et al., 2021a), which may not effectively reflect current emission sources, industrial developments, or recent advancements in pollution control.

In response to the need for more accurate air quality simulations, which are essential for proceeding with further analyses based on their outcomes, extensive efforts have been made to refine emissions inventories across Asia. Traditional bottom-up methods, such as ground surveys and industrial reporting, however, can be time-consuming and often outpaced by rapidly evolving emissions patterns, making them less reflective of contemporary emissions activities (Placet et al., 2000; Rypdal & Winiwarter, 2001; Li et al., 2021; Smith et al., 2022). To address this, satellite observation data have widely been used to update emissions inventories in a top-down manner, taking advantage of the instruments' broader geographic coverage. Instruments aboard sun-synchronous low Earth orbit satellites, such as the Ozone Monitoring Instrument (OMI) and the TROPOspheric Monitoring Instrument (TROPOMI), have been particularly effective in constraining the extent of air pollutant emissions. More recently, the Geostationary Environment Monitoring Spectrometer (GEMS) has further enhanced this process by capturing daytime variations in pollutant loadings, such as tropospheric nitrogen dioxide ($NO_2$) columns, offering unprecedented insights into the diurnal behaviors of pollutants across Asia. Leveraging the top-down information, a number of studies have successfully refined Asia's emissions inventories, achieving substantial improvements in CTM-based simulation accuracy (Itahashi et al., 2012; Yumimoto et al., 2014; Goldberg et al., 2019; Souri et al., 2020; Li et al., 2021b; Jung et al., 2022; Son et al., 2022; Feng et al., 2023; Mun et al., 2023; Park et al., 2023; Momeni et al., 2024; Park et al., 2024). However, only a few have extended their scope to utilize these refined simulations for further analyses of cross-regional pollutant dynamics across multiple seasons; updating emissions inventories itself is already a resource-intensive process, demanding a series of forward model runs and iterative adjustments. For example, Souri et al. (2020) refined $NO_x$ and VOC emissions across East Asia during the 2016 KORUS-AQ campaign period, through analytical inversion using satellite data from OMI and Ozone Mapping and Profiler Suite Nadir Mapper (OMPS-NM) and CMAQ. The inversion led to significant reductions in $NO_x$ emissions by 22%-41% in China, Taiwan, and Malaysia, while Korea and Japan showed increases by 9%-12%. VOC emissions over the North China Plain were adjusted upward by 25%, a significant increase compared to the previously reported 5% since 2010. This involved shifts in chemical regimes across East Asia, with regions transitioning between $NO_x$-sensitive and VOC-sensitive conditions, providing more recent insights into ozone formation risks across different regions. Similarly, Jung et al. (2022) refined East Asia's $NO_x$ emissions inventory for spring 2019 through Bayesian inversion using TROPOMI data and CMAQ-ISAM. The use of more up-to-date emissions substantially reduced model biases in simulating $NO_2$, ozone, and $PM_{2.5}$ concentrations, revealing that the prior emissions were underestimating the contributions of transboundary pollutants. $NO_x$ emissions from neighboring regions contributed 22.96%–35.24% to local $NO_x$ budgets and 24.23%–42.26% to ozone budgets in both China and Korea, reaffirming the critical role of anticyclonic systems over the Yellow Sea in driving pollutant transport. Beyond these satellite-based studies, several others successfully performed source apportionment of East Asia's air pollutants using their emissions inventories

updated for recent years (Choi et al., 2019; Huang et al., 2021; Feng et al., 2022; Zhang et al., 2023). For example, Choi et al. (2019)'s study utilized in-situ observation data from six monitoring sites across Korea to update the inventoried extent of $PM_{2.5}$ precursor emissions, including $NO_x$, $SO_2$, ammonia, organic carbon, and black carbon, during the 2016 KORUS-AQ campaign, using the GEOS-Chem model and its adjoint. Under stagnant meteorological conditions, local emissions were the dominant source of surface $PM_{2.5}$ concentrations in Korea, contributing up to 57% of the total mass. During extreme pollution episodes, transboundary emissions from China became the primary contributor, accounting for up to 68% of $PM_{2.5}$ concentrations. Despite these efforts, there is still a need for more rigorous efforts in this domain to better explore regional pollution dynamics. Particularly, the recurrent pollutant patterns during East Asia's winter and spring seasons deserve updated perspectives on the complex interplay between evolving synoptic systems and pollutant transport dynamics, which significantly affect air quality across the region.

Leveraging top-down estimates of $NO_x$ emissions informed by GEMS tropospheric $NO_2$ columns, followed by improved accuracy in CTM simulations, our study aims to conduct a comprehensive source apportionment of East Asia's $NO_y$ concentrations during the winter-spring transition in 2022. First, we applied diurnal updates to the inventoried extent of $NO_x$ emissions using GEMS $NO_2$ columns as top-down constraints in our Bayesian inversion, enabling CMAQ to more accurately simulate $NO_y$ concentrations. Then, using CMAQ-ISAM, we quantified the local and transboundary contributions of $NO_x$ emissions to $NO_y$ concentrations across five major $NO_x$ source regions of East Asia during the period from January to May 2022. By capturing the response of pollutants to evolving seasonal dynamics, we assessed the source-receptor interplays between the regions, aiming to provide more up-to-date insights into the broader cross-regional pollution transport dynamics.

## 2 Materials and Methods

### 2.1 Models

Meteorology governs the dispersion and transport of air pollutants, making it a critical factor in CTM simulations. We used the Weather Research and Forecasting (WRF) model 3.8.1 (Skamarock et al., 2008) to simulate meteorological fields over the modeling domain (Figure 1). The simulation spanned from January 1 to May 31, 2022, covering East Asia's winter and spring seasons. We simulated hourly meteorological fields over a 320 × 320 grid with 35 vertical layers at a spatial resolution of 27 km and then converted them into a CMAQ-compatible format using the Meteorology-Chemistry Interface Processor (MCIP). We used the Morrison two-moment scheme for microphysics (Morrison et al., 2009), the Rapid Radiative Transfer Model for GCMs (RRTMG) for longwave and shortwave radiation (Clough et al., 2005; Iacono et al., 2008), and the Pleim-Xiu land surface and surface layer models (Xiu and Pleim, 2001; Pleim, 2006). Planetary boundary layer processes

were simulated using the ACM2 model (Pleim, 2007a; Pleim, 2007b), and cumulus parameterization was handled by the
Kain-Fritsch scheme (Kain, 2004). We applied the Four-Dimensional Data Assimilation (FDDA) grid-nudging option (Jeon
et al., 2015) for meteorological inputs. Initial and boundary conditions were derived from the National Centers for
Environmental Prediction (NCEP) FNL operational model global tropospheric analysis.

Using the WRF-simulated meteorology and established emissions as inputs, CMAQ simulates the behavior and distribution
of pollutants in the atmosphere in a three-dimensional manner. We employed two different versions of CMAQ: CMAQ 5.2
with its Decoupled Direct Method in Three Dimensions (CMAQ DDM-3D) as a forward model in the emission adjustment
process, and CMAQ 5.3.2 with its ISAM for performing source apportionment. CMAQ DDM-3D calculates the first-order
coefficients that represent locally semi-normalized sensitivities of modeled pollutant concentrations to changes in relevant
emissions input (Napelenok et al., 2006). CMAQ-ISAM tags pollutants emitted from user-defined source regions, and then
tracks them through atmospheric processes such as advection, chemical transformation, and deposition (Kwok et al., 2015).
This allows for quantifying the contribution of specific emission sources to pollutant loadings at receptor locations across the
modeling domain, offering detailed insights into source attribution. Building upon previous studies conducted in Asia (Jung
et al., 2022; Park et al., 2023; Park et al., 2024), we employed CMAQ configurations that have been validated in comparable
contexts across the region. Using CMAQ DDM-3D, we simulated hourly $NO_2$ concentrations over a $300 \times 300$ grid and
obtained their corresponding sensitivities to $NO_x$ emissions, which were used for our Bayesian inversion to constrain the
inventoried extent of $NO_x$ emissions (details in Sections 2.2 and 2.4). We used the YAMO scheme and the WRF omega
formula for solving horizontal advection and vertical advection, respectively. Horizontal diffusion was modeled using the
multiscale approach, and vertical diffusion was represented with the ACM2 vertical diffusion scheme (Pleim, 2007a; Pleim,
2007b). Gas-phase chemistry was solved using the Carbon Bond 05 (CB05) mechanisms for CMAQ DDM-3D and CB6 for
CMAQ-ISAM. Aerosol processes were modeled using the AERO6 module, and dry deposition was estimated using the
M3Dry scheme (Pleim, 2007b). We used static boundary conditions during the entire simulation period. Using CMAQ-
ISAM, we quantified the local and transboundary contributions of $NO_x$ emissions to $NO_y$ concentrations among five selected
regions (Figure 1), including Korea, Northeast China, North China, East China, and South Central China. During discussions
for Korea (later in Section 3), we focused on the SMA, the country's economic hub, where dense traffic activity contribute to
severe air pollution (Figure 1) (Park & Lee, 2020). North Korea, despite its close proximity to these regions and the potential
impact of its emissions on neighboring regions' air quality, was excluded from our study. This was primarily due to the
uncertainty in North Korea's emissions, which warrants a dedicated study of its own. Global emissions inventories like
EDGAR rely on accurate energy usage data to estimate pollutant emissions, and the lack of reliable input data for North
Korea made it impractical to include as a separate source region in this study. Our focus was on total $NO_y$ concentrations
within the PBL at receptor regions rather than surface $NO_x$ or $NO_2$ concentrations due to the latter's short atmospheric
lifetimes, which limit their long-range transport. The PBL facilitates the most efficient mixing and transport across regions,
making it an ideal layer for assessing cross-regional behaviors of pollutants (Li et al., 2017b). However, it is important to

acknowledge that substantial pollutant transport also occurs in the free troposphere beyond the PBL, where stronger winds facilitate long-range movement of pollutants. Our study specifically focuses on the PBL to assess cross-regional pollutant behaviors, as this layer directly influences surface air quality, the modeled estimates of which can be evaluated with station measurements (detailed in Section 2.5), and human health associated. We quantified the extent to which $NO_y$ remained near local sources versus how much was transported to neighboring receptor regions during the winter-spring months. Note that we used the summation of $NO_x$, nitric acid ($HNO_3$), nitrous acid (HONO), and peroxyacetyl nitrate (PAN) to represent $NO_y$, capturing the major reactive nitrogen species that contribute to total $NO_y$ concentrations during the simulations. Both the WRF and CMAQ simulations began with a 10-day spin-up from December 22 to December 31, 2021. Further technical details of our modeling setup are listed in Table S1.

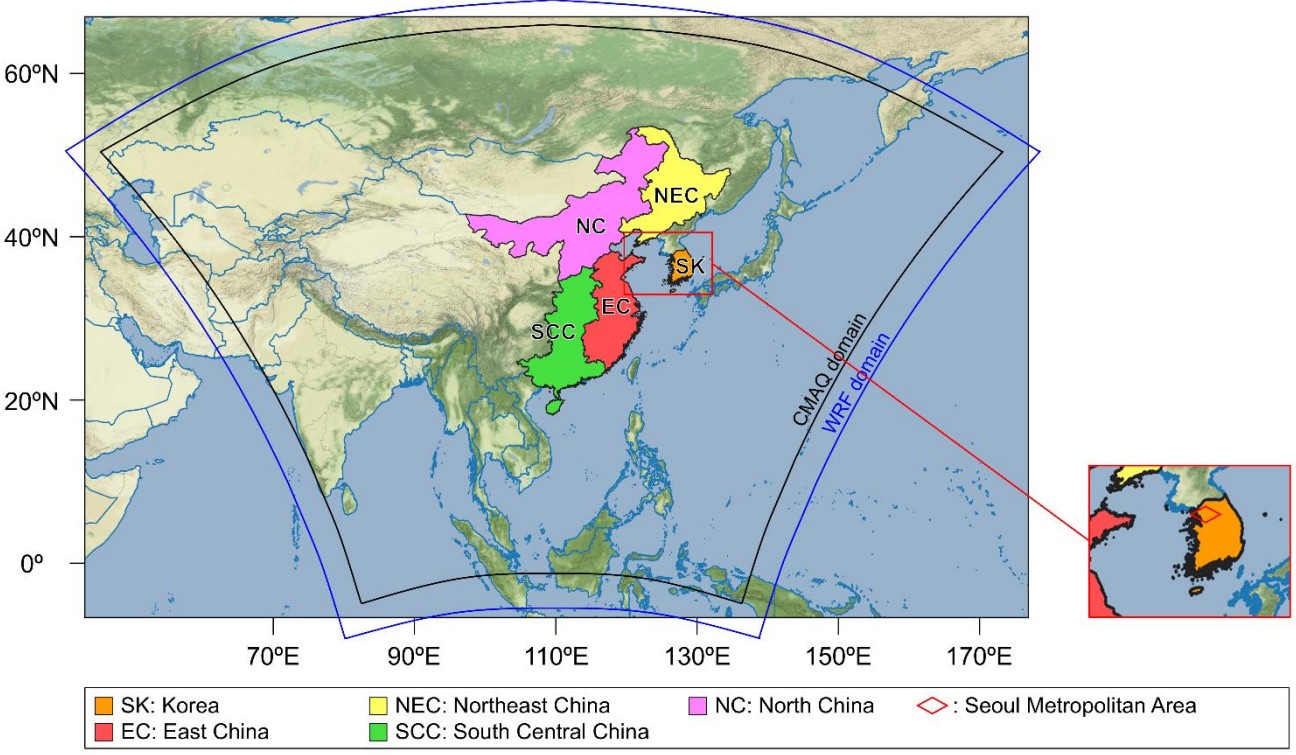

**Figure 1:** Modeling domain and five source apportionment regions across East Asia.

**2.2 Emissions**

Emissions inventories provide CTMs with spatiotemporally resolved information on the extent of air pollutant emissions, enabling the simulation of their behavior and resulting concentrations in the atmosphere. We prepared anthropogenic emissions over the modeling domain using the Emissions Database for Global Atmospheric Research (EDGAR) 6.1 (Crippa

et al., 2020), which offers annual data (base year: 2018) for greenhouse gas and air pollutant emissions at a 0.1° spatial resolution. We processed these emissions into a CMAQ-compatible format using the Sparse Matrix Operator Kernel Emissions (SMOKE) 4.7 modeling system (Houyoux et al., 2000). This process involved re-gridding the emissions into a 27 km resolution and allocating the annual lumped emissions into hourly speciated emissions for the period from January 1 to May 31, 2022, while accounting for time zones and weekday weekend profiles that vary across geographical locations. It is noteworthy that, unlike our previous study (Park et al., 2023) and other literatures that inspired our methodology (Souri et al., 2020; Jung et al., 2022), we did not use the 2016 KORUS-AQ emissions inventory (version 5) developed by Woo et al. (2020). This inventory, widely adopted for Asia-focused studies, was delicately built upon a combination of multiple regional surveys, providing comprehensive representations of emissions patterns across Asia. However, since the KORUS-AQ inventory provides the inventoried extent of pollutant emissions as monthly and yearly totals and given the requirements of our inversion method (detailed in Section 2.4) to update Asia's emissions inventory on an hourly basis, we used the EDGAR inventory instead. The EDGAR database offers region-specific hourly emissions profiles, allowing us to avoid introducing additional uncertainties associated with the temporal allocation of monthly emissions into hourly emissions. We prepared biogenic and biomass burning emissions using the Model of Emissions of Gases and Aerosols from Nature (MEGAN) 3.0 (Guenther et al., 2018) and the Fire Inventory from the National Center for Atmospheric Research (NCAR) (FINN) 1.5 (Wiedinmyer et al., 2011). MEGAN estimates the extent of gases and aerosol emissions from terrestrial ecosystems based on vegetation responses to meteorological conditions. We obtained hourly biogenic emissions at a 27 km resolution using the WRF-simulated meteorological fields and Leaf Area Index (LAI) averaged over vegetative surfaces (referred to as LAIv) as an input, which can be calculated by dividing grid-specific LAI by the fraction of each grid cell covered by vegetation (Guenther, 2006). To calculate LAIv, we used the Reprocessed Moderate Resolution Imaging Spectroradiometer (MODIS) Version 6 LAI product (Yuan et al., 2011) and the Visible Infrared Imaging Radiometer Suite (VIIRS) global Green Vegetation Fraction product (Jiang et al., 2016), following the method established in previous studies (Jung et al., 2021; Park et al., 2022). This ensures that MEGAN estimates biogenic emissions based on more contemporary information on vegetative surfaces over the modeling domain. FINN provides emissions from open biomass burning events, such as wildfires, agricultural fires, and prescribed burning, based on satellite observation data and fuel load parameters. We obtained hourly biomass burning emissions at a 27 km resolution, using the Fortran-based gridding program (fire_emis) provided by NCAR, which spatiotemporally allocated FINN emissions into our modeling grid. We merged these anthropogenic, biogenic, and biomass-burning emissions to prepare a comprehensive emissions input for CMAQ (hereafter referred to as the a priori emissions). Note that our use of two different CMAQ versions required species mapping, as EDGAR emissions were provided in the CB05 mechanism, which CMAQ DDM-3D can digest, but CMAQ-ISAM requires those in CB6. We converted the chemical species from CB05 to CB6 using the species mapping method described in a previous study (Collet et al., 2018).

**2.3 Satellite data**

GEMS is the first ultraviolet-visible geostationary instrument capable of capturing diurnal profiles of both gaseous pollutants and aerosols across the Asia-Pacific region, covering longitudes from 75° E to 145° E and latitudes from 5° S to 45° N (Choi et al., 2018). We used GEMS Level 2 $NO_2$ product (version 2.0) to obtain a top-down overview of $NO_2$ loadings across the modeling domain. This product, including observations from November 2020 to the present, provides 6 to 10 consecutive snapshots of $NO_2$ column densities at hourly intervals during the daytime, at a spatial resolution of 3.5 km × 8 km. For

clarification, GEMS provided 6 observations per day from 00:45 to 05:45 UTC in January, 7 observations from 00:45 to 06:45 UTC in February, 8 observations from 23:45 to 06:45 UTC in March, and 10 observations from 22:45 to 07:45 UTC in April and March.

We used tropospheric $NO_2$ columns observed from January 1 to May 31, 2022, as top-down references for constraining the a priori emissions. In addition to the $NO_2$ columns, we incorporated several other variables during the inversion process (see

Section 2.4), including the averaging kernel, cloud fraction, data quality flags, and root mean square error. We also used model-derived variables from the GEMS Level 2 data, including tropospheric and stratospheric air mass factors (AMFs), a priori tropospheric $NO_2$ profile, and tropospheric pressure profile from the WRF model coupled with Chemistry (WRF-Chem) 3.9.1 (NIER, 2020). To ensure consistency in the vertical distribution assumptions between GEMS tropospheric $NO_2$ columns and CMAQ-simulated tropospheric $NO_2$ columns, we recalculated the AMFs using the vertical $NO_2$ profiles

simulated by CMAQ. These AMFs were then used to adjust the GEMS-retrieved $NO_2$ columns, aligning them with CMAQ's vertical profiles. This process involves interpreting the satellite retrievals using CMAQ's vertical $NO_2$ profiles instead of the original GEMS a priori vertical profiles, which were based on WRF-Chem. This adjustment ensures that both the $NO_2$ columns are interpreted consistently with the model vertical structure, thereby mitigating biases introduced by differences in the initial assumptions about the vertical distribution of $NO_2$. This approach builds upon methods established in previous

inverse modeling studies that directly compared OMI- and TROPOMI-derived $NO_2$ columns with CMAQ simulations (Souri et al., 2016; Souri et al., 2017; Souri et al., 2020; Jung et al., 2022a; Jung et al., 2022b). To ensure data quality, we used pixels with a quality flag of 0 bits (good sample) and cloud fractions below 0.3; note that since the Level 2 data version 2.0 quit employing the OMI climatology thereby deserves further validation efforts through retrieval studies, we excluded the cloudy scenes which could lead to inaccurate AMF references.

**2.4 Top-down approach to constrain $NO_x$ emissions**

The extent of $NO_x$ emissions is not directly measurable through GEMS's observations, which instead capture $NO_2$ column densities. While these are closely related to $NO_x$ emissions, they do not provide direct quantitative measurements of the emissions themselves. Therefore, to establish quantitative constraints on $NO_x$ emissions and obtain the updated estimates accordingly, we employed a Bayesian approach for inverse modeling, which is suited for solving problems that are not

grossly nonlinear (Rodgers, 2000).

Given the short atmospheric lifetime of $NO_2$, our approach assumes a local, linear relationship between $NO_2$ columns and $NO_x$ emissions. This assumption, widely adopted in earlier inverse modeling studies (Martin et al., 2003; Souri et al., 2016, 2018, 2020; Jung et al., 2022a, 2022b; Park et al., 2023, 2024), is based on the understanding that satellite-observed $NO_2$ columns are most likely to reflect recent, localized $NO_x$ emissions due to $NO_2$'s short atmospheric lifetime. Furthermore, the linearity simplifies the inversion process by directly relating observed column densities to emissions, avoiding the need for computationally intensive modeling of nonlinear processes. However, it is noteworthy that the observed $NO_2$ columns at any given time are influenced not only by current $NO_x$ emissions but also by $NO_2$ remaining from previous hours. Additionally, nighttime chemical reactions involving ozone and hydroxyl radicals (OH) can introduce nonlinearity between $NO_x$ emissions and $NO_2$ concentrations, which is a complexity beyond the scope of our study. Assuming that uncertainties in observations and emissions follow a Gaussian distribution, our approach aimed to derive the most probable estimate of a posteriori $NO_x$ emissions by integrating prior knowledge (a priori emissions) and top-down observational constraints. This involves minimizing the cost function derived from Bayes's theorem, as shown in Eq. 1 (Rodgers, 2000).

$$J(x) = \frac{1}{2}(y - Fx)^T S_o^{-1}(y - Fx) + \frac{1}{2}(x - x_a)^T S_e^{-1}(x - x_a) \qquad (1)$$

This process determined the a posteriori emissions $x$ for each grid cell, given multiple inputs: the a priori emissions $x_a$, observation constraints $y$ (hourly GEMS $NO_2$ columns), and CMAQ-simulated $NO_2$ columns $F$. Due to the 15-minute offset in the availability of GEMS Level 2 products (from 22:45 to 07:45 UTC), we aligned the observations to the nearest subsequent hour (e.g., GEMS data at 04:45 UTC were used to constrain emissions at 05:00 UTC). We assumed emissions uncertainties $S_e$ at 50%, 200%, and 100% for anthropogenic, biogenic, and biomass burning emissions, respectively, based on previous modeling studies conducted across Asia (Souri et al., 2020; Jung et al., 2022; Park et al., 2023; Park et al., 2024). Observation uncertainty $S_o$ was sourced from the GEMS Level 2 data.

Once the first derivative of the cost function reached its minimum, we applied the Gauss-Newton method to iteratively refine the emissions estimate, as shown in Eq. 2. This process involved adjusting the estimate $x$ (with each iteration noted as $i$; $i =$ 4 in January, February, and $i = 3$ in March, April, and May), gradually progressing towards a converged solution. The Jacobian matrix $K$, which represents the sensitivity between $NO_x$ emissions and $NO_2$ concentrations, was calculated by CMAQ DDM-3D at the beginning of the simulations and remained fixed throughout the inversion process. Meanwhile, the forward model $F$ was updated with each iteration, guiding the inversion toward reducing discrepancies between the observed and modeled $NO_2$ columns.

$$\hat{x}_{i+1} = x_a + S_e K_i^T (K_i S_e K_i^T + S_o)^{-1}[y - Kx_i + K_i(x_i - x_a)] \qquad (2)$$

The inversion was applied whenever the top-down constraints were available, allowing us to constrain hourly $NO_x$ emissions during GEMS's daylight retrieval hours, while keeping nighttime emissions unchanged. We chose not to adjust nighttime emissions primarily due to the absence of observational reference during these hours and also to isolate the daily emission cycle, allowing the model to "pause" and mitigate carry-over effects from the previous day's emissions (Park et al., 2024). However, a limitation of this approach is that it does not improve the representation of nighttime emissions, leaving these unadjusted due to the absence of observational constraints. Consequently, any model biases associated with nighttime emissions remain unresolved.

## 2.5 Station measurements for model evaluation

Before using the WRF-simulated meteorological fields as input for CMAQ, we evaluated their accuracy against ground-based measurements at weather stations operated by the Korean Meteorological Administration. We used hourly measurements of 2 m air temperature and 10 m wind U and V components from 95 stations, collected for the period from January 1 to May 31, 2022. The modeled meteorology showed fair agreement with station measurements (Figure S1), with Pearson's correlation coefficients (R) ranging from 0.89 to 0.98 and Index of Agreement (IOA) values between 0.83 and 0.98.

To evaluate the accuracy of CMAQ simulations, we used hourly surface $NO_2$ and $PM_{2.5}$ concentrations observed at ground-based monitoring stations in Korea and China during the period from January 1 to May 31, 2022, sourced from Korea's Ministry of Environment (AirKorea) and China's Ministry of Ecology and Environment (MEE). To ensure the quality of AirKorea measurements, from an original count of 515 stations, we excluded those with more than 50% missing data during the validation period (Park et al., 2022; Park et al., 2023), which resulted in a 9.68% data loss and retaining 465 stations. To ensure the quality of MEE measurements, we applied data filtering methods to data from 185 national control points, consisting of 20 sites in Northeast China, 38 sites in North China, 78 sites in East China, and 49 sites in South Central China. These control points are strategically distributed across China and managed by its central government to ensure consistent and reliable air quality measurements (Liu et al., 2021a; Liu et al., 2022). We excluded negative values and duplicate records (> 4 consecutive repeats) caused by equipment failures, following the data-filtering methods in previous studies (Rohde, 2015; Silver, 2018; Zhai, 2019). This resulted in a decrease in the number of data points by 0.41% for $NO_2$ and 0.38% for $PM_{2.5}$, respectively. Note that we converted MEE's $NO_2$ measurements from its native unit $\mu g/m^3$ to ppb.

## 3 Results and discussion

### 3.1 Model evaluation

Prior to proceed with source apportionment, we first evaluated the accuracy of our CMAQ simulations across East Asia. We compared the observed and modeled tropospheric $NO_2$ columns from using the a priori and a posteriori $NO_x$ emissions, in order to evaluate the improvement in the model's performances. Figure 2 shows monthly averages of hourly daytime $NO_2$ columns during the period from January to May 2022. The prior model generally underestimated $NO_2$ columns across North China, Northeast China, and the northern half of East China, and the SMA of Korea (Figures 2a and 2b). The extent of underestimation was particularly pronounced in North China, failing to capture highly polluted areas observed throughout the simulation period. To a lesser extent, the model sporadically overestimated the columns across South Central China, the southern half of East China, and the rest of Korea except the SMA.

After the inversion, there were substantial adjustments to the extent of daytime $NO_x$ emissions across the regions (Figure S4), seemingly counterbalancing the earlier model under- and overestimation. During the winter-spring months, on average, there were increases in the emissions by 50.12% in Korea, 30.86% in Northeast China, 78.63% in North China, 20.76% in East China, and 2.6% in South Central China (Table S2). Overall, these adjustments led to a closer alignment between the observed and modeled $NO_2$ columns (Figure 2c). For example, in regions such as North China, including Beijing, and parts of Northeast China, such as Shenyang, $NO_2$ columns substantially increased by a factor of approximately 1.2 to 2.0 during winter months (January and February). Similarly, the SMA of Korea experienced moderate increases by a factor of 1.2 to 1.5, during the same period. In contrast, in South Central China, $NO_2$ columns showed a mix of increases and decreases depending on the month. In May, for instance, $NO_2$ columns in South Central China decreased by a factor of about 1.3, compensating for the model's earlier overestimation. In addition, we noticed some posterior overcompensation in the modeled columns, shown by some overestimated values across North and South Central China and Korea in February and March, which could have been caused by our use of the Bayesian approach, often regarded as a simple inverse modeling method which cannot fully resolve the non-locality of air pollutants (Park et al., 2024). Nonetheless, overall, our use of the a posteriori $NO_x$ emissions yielded a more accurate spatial representation of $NO_2$ concentrations across the regions during the winter-spring transition, demonstrating its effectiveness in refining emissions inventories and thereby improving model accuracy.

## Tropospheric NO₂ columns observed and modeled in Asia

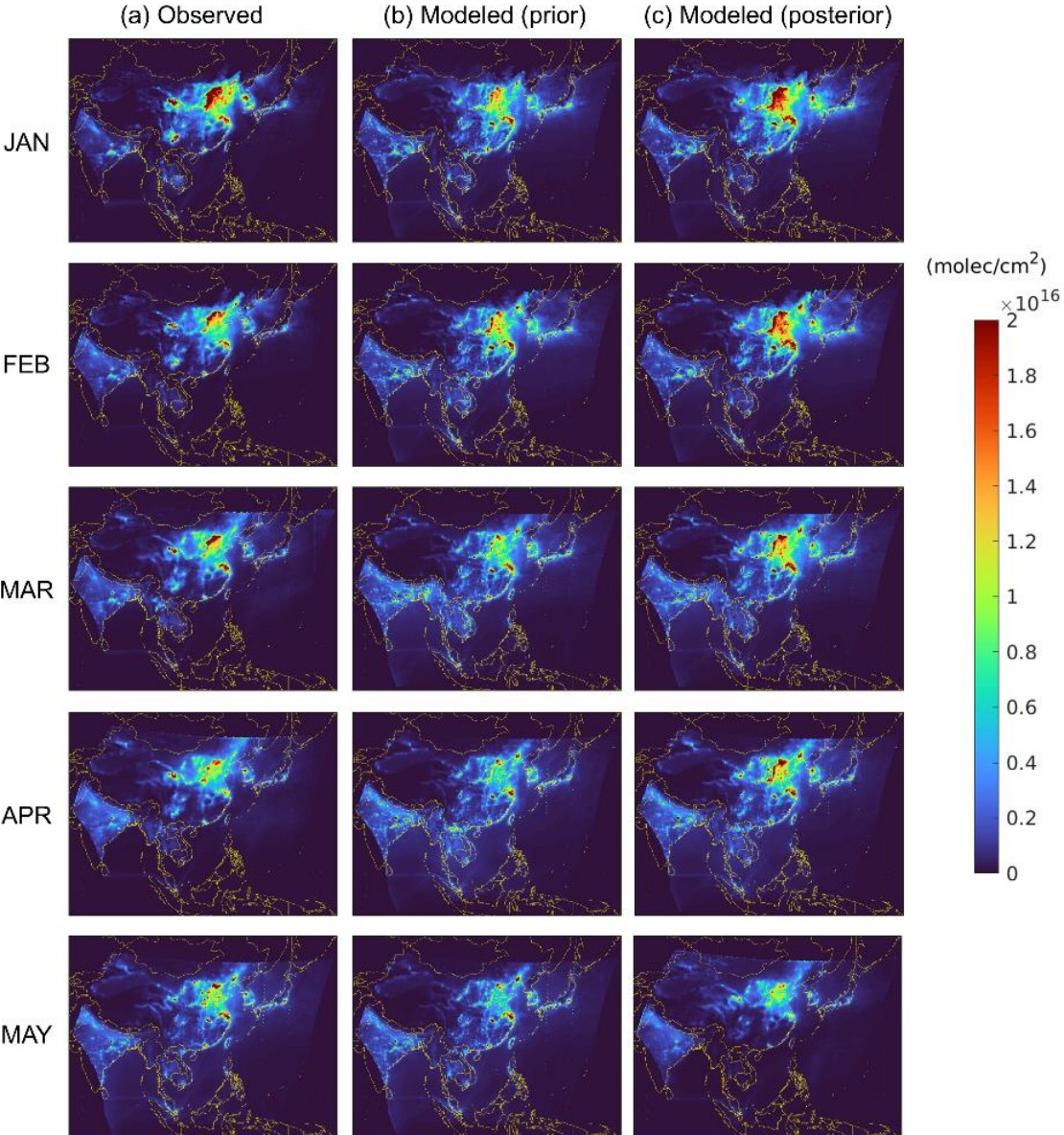

**Figure 2:** Averages of hourly tropospheric NO₂ columns (molecules/cm²) observed and modeled during daylight hours (GEMS retrieval hours) in each month from January to May 2022. (a) GEMS tropospheric NO₂ columns, (b) modeled NO₂ columns using the a priori emissions, (c) modeled NO₂ columns using the a posteriori emissions. Note that we excluded the modeled columns that do not correspond with the GEMS's retrieval times.

The increases in NO$_x$ emissions reduced the extent of model biases in simulating daytime surface NO$_2$ concentrations from -32.75% to -13.01% in Korea and from -10.26% to -3.04% in China on average (from -39.06% to -9.40% in Northeast China, from -2.78% to 2.76% in North China, from -9.65% to -3.32% in East China, and from -12.41% to -10.92% in South Central China) (Figure 3; Table 1). This led to closer alignment between the modeled and observed concentrations during the months, showing improvements in R from 0.67 to 0.71 and IOA from 0.71 to 0.82 in Korea, and R from 0.65 to 0.76 and IOA from 0.79 to 0.86 in China. Despite the improvements, the extent of the prior underestimation still remained the largest in January, gradually decreasing as the months progressed to May. This suggests a potential underrepresentation of NO$_x$ emissions in the a priori inventory during the colder months, possibly due to the use of the global emissions database, which may not fully capture localized, high energy usage in those periods. There were instances of overcompensation after the inversion, such as in March and April for North China, as well as in April for East China and May for South Central China, where the prior model's underestimations turned into overestimations (Table 1). This could be attributed to the carry-over effect of NO$_2$ transport from neighboring grid cells, which our Bayesian inversion, which assumes a strictly local relationship between emissions and concentrations, cannot fully constrain (Park et al., 2024). Nevertheless, the reductions in mean absolute errors across these regions suggest an overall improvement in simulation accuracy.

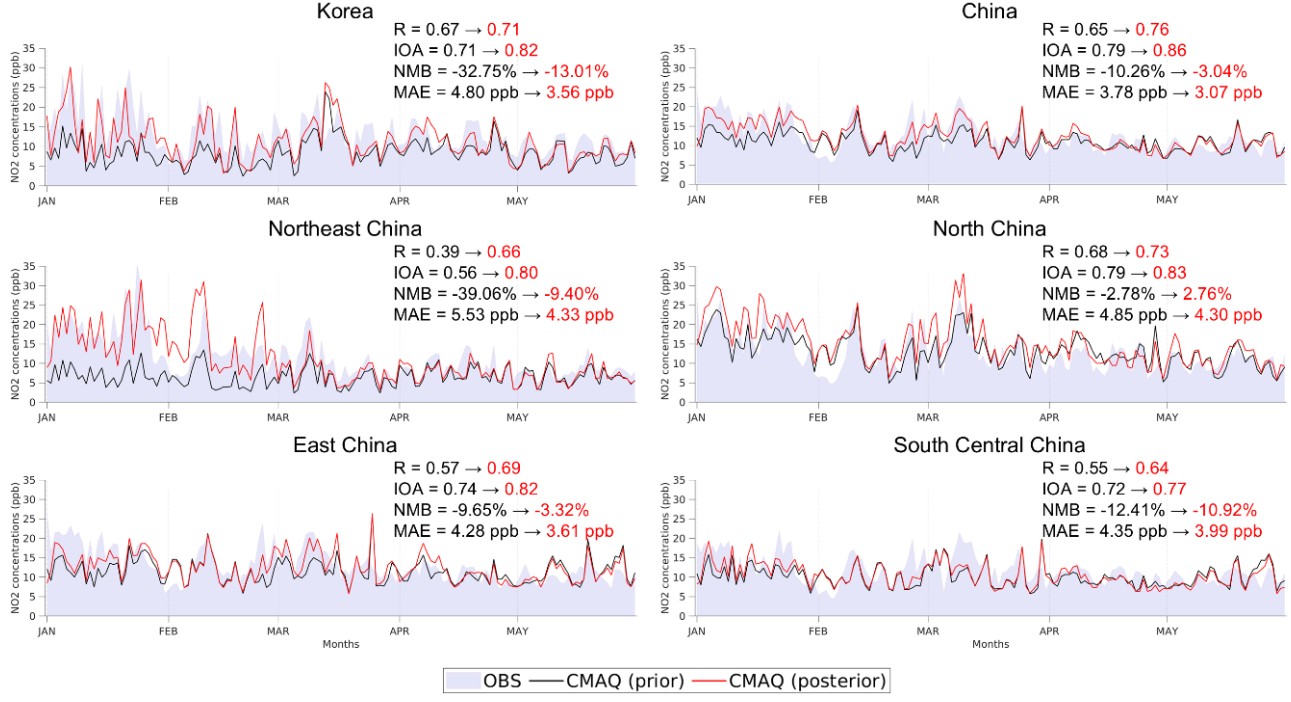

**Figure 3:** Daytime mean surface NO₂ concentrations observed and modeled at ground-based measurement sites within each of the five source apportionment regions across East Asia during the period from January 1 to May 31, 2022. OBS: observed concentrations, CMAQ (prior): modeled concentrations using the a priori NO$_x$ emissions, CMAQ (posterior): modeled concentrations using the a posteriori NO$_x$ emissions. Arrows indicate the changes in metrics from the prior model to the posterior model. R: Pearson's correlation coefficient, IOA: Index of Agreement, NMB: normalized mean bias (%), MAE: mean absolute error (ppb).

**Table 1:** Descriptive statistics comparing observed and modeled hourly surface $NO_2$ concentrations at ground-based measurement sites within each of the five source apportionment regions across East Asia during the period from January to May 2022. Prior: modeled concentrations using the a priori $NO_x$ emissions, Posterior: modeled concentrations using the a posteriori $NO_x$ emissions. R: Pearson's correlation coefficient, IOA: Index of Agreement, NMB: normalized mean bias (%), MAE: mean absolute error (ppb).

| | | | January | February | March | April | May | 5-month |
|---|---|---|---|---|---|---|---|---|
| Korea | Prior | R | 0.89 | 0.88 | 0.63 | 0.73 | 0.82 | 0.67 |
| | | IOA | 0.60 | 0.63 | 0.74 | 0.80 | 0.79 | 0.71 |
| | | NMB | -50.98 | -48.58 | -21.39 | -20.64 | -27.75 | -32.75 |
| | | MAE | 8.89 | 5.97 | 4.64 | 3.43 | 3.04 | 4.80 |
| | Posterior | R | 0.55 | 0.80 | 0.64 | 0.71 | 0.77 | 0.71 |
| | | IOA | 0.71 | 0.85 | 0.79 | 0.83 | 0.82 | 0.82 |
| | | NMB | -17.37 | -22.14 | -4.17 | -6.15 | -18.84 | -13.01 |
| | | MAE | 5.85 | 3.65 | 4.02 | 2.93 | 2.39 | 3.56 |
| Northeast China | Prior | R | 0.59 | 0.51 | 0.72 | 0.41 | 0.26 | 0.39 |
| | | IOA | 0.46 | 0.55 | 0.69 | 0.63 | 0.52 | 0.56 |
| | | NMB | -62.23 | -50.37 | -39.57 | -18.70 | -17.10 | -39.06 |
| | | MAE | 11.64 | 6.46 | 4.40 | 4.01 | 3.65 | 5.53 |
| | Posterior | R | 0.56 | 0.56 | 0.72 | 0.38 | 0.23 | 0.66 |
| | | IOA | 0.73 | 0.72 | 0.79 | 0.63 | 0.51 | 0.80 |
| | | NMB | -1.57 | 1.15 | -25.94 | -10.67 | -12.35 | -9.40 |
| | | MAE | 5.67 | 5.16 | 3.58 | 4.03 | 3.91 | 4.33 |
| North China | Prior | R | 0.67 | 0.61 | 0.75 | 0.66 | 0.69 | 0.68 |
| | | IOA | 0.79 | 0.78 | 0.84 | 0.72 | 0.76 | 0.79 |
| | | NMB | -11.37 | -3.15 | -2.49 | 10.93 | -8.03 | -2.78 |
| | | MAE | 5.28 | 5.27 | 4.53 | 5.27 | 4.18 | 4.85 |
| | Posterior | R | 0.58 | 0.63 | 0.78 | 0.72 | 0.80 | 0.73 |
| | | IOA | 0.74 | 0.79 | 0.86 | 0.78 | 0.86 | 0.83 |
| | | NMB | -0.33 | -1.47 | 6.64 | 15.33 | -7.91 | 2.76 |
| | | MAE | 5.82 | 4.72 | 4.28 | 4.40 | 3.04 | 4.30 |
| East China | Prior | R | 0.62 | 0.49 | 0.60 | 0.67 | 0.63 | 0.57 |
| | | IOA | 0.67 | 0.69 | 0.74 | 0.75 | 0.77 | 0.74 |
| | | NMB | -27.63 | -13.79 | -13.88 | -1.71 | 8.38 | -9.65 |
| | | MAE | 5.41 | 4.33 | 4.77 | 3.76 | 3.69 | 4.28 |
| | Posterior | R | 0.58 | 0.68 | 0.71 | 0.78 | 0.71 | 0.69 |
| | | IOA | 0.73 | 0.81 | 0.81 | 0.81 | 0.82 | 0.82 |
| | | NMB | -16.19 | -8.86 | -0.55 | 4.00 | 3.67 | -3.32 |
| | | MAE | 4.29 | 3.41 | 3.93 | 3.46 | 3.23 | 3.61 |
| South Central China | Prior | R | 0.65 | 0.48 | 0.51 | 0.64 | 0.63 | 0.55 |
| | | IOA | 0.73 | 0.69 | 0.67 | 0.71 | 0.74 | 0.72 |
| | | NMB | -23.43 | -12.04 | -16.33 | -13.00 | 1.79 | -12.41 |
| | | MAE | 4.25 | 4.04 | 5.64 | 4.32 | 3.61 | 4.35 |
| | Posterior | R | 0.63 | 0.57 | 0.57 | 0.72 | 0.68 | 0.64 |
| | | IOA | 0.77 | 0.75 | 0.71 | 0.76 | 0.78 | 0.77 |
| | | NMB | -12.17 | -9.84 | -14.60 | -12.83 | -4.75 | -10.92 |
| | | MAE | 3.90 | 3.68 | 5.16 | 3.91 | 3.38 | 3.99 |
| China | Prior | R | 0.72 | 0.61 | 0.68 | 0.69 | 0.68 | 0.65 |

| (Northeast, North, East, and South Central) | | IOA | 0.73 | 0.76 | 0.80 | 0.76 | 0.80 | 0.79 |
| | | NMB | -25.65 | -13.50 | -12.79 | -2.00 | 2.16 | -10.26 |
| | | MAE | 4.72 | 3.81 | 4.14 | 3.58 | 3.09 | 3.78 |
| | Posterior | R | 0.67 | 0.73 | 0.77 | 0.78 | 0.76 | 0.76 |
| | | IOA | 0.81 | 0.85 | 0.86 | 0.83 | 0.85 | 0.86 |
| | | NMB | -8.44 | -5.13 | -3.20 | 2.26 | -1.15 | -3.04 |
| | | MAE | 3.67 | 3.08 | 3.35 | 2.93 | 2.62 | 3.07 |

## 3.2 Source apportionment

Upon improving model accuracy, we assessed the contributions of $NO_x$ emissions from local and outside sources to total $NO_y$ concentrations in each of the five regions of East Asia during the winter-spring transition. $NO_y$ concentrations were generally higher near the source regions throughout the months, with notable transboundary transport extending to downwind regions (Figure 4). While local contributions remained substantial across all regions, a clear decreasing trend was seen as the season progressed. Local sources' contributions were greater during the winter months compared to those from transboundary sources, whereas spring months (March, April, and May) showed a marked increase in transboundary contributions across the regions (Figure 4; Table 2). In January, local contributions were 31.50%, 42.62%, 37.42%, 33.69%, and 31.79% in Korea, Northeast China, North China, East China, and South Central China, respectively. By May, these decreased to 23.24%, 26.29%, 25.28%, 29.05%, and 30.06%. Meanwhile, transboundary contributions steadily increased as the months progressed. In January, transboundary contributions were 27.16%, 16.17%, 16.79%, 30.46%, and 33.39% in Korea, Northeast China, North China, East China, and South Central China, respectively. By May, these increased to 36.89%, 32.46%, 27.64%, 35.57%, and 35.70%. The decreases in local contributions can be partially attributed to reduced energy use and subsequent decline in anthropogenic emissions as the seasons transitioned to warmer months. However, this does not fully explain the concurrent increase in transboundary contributions, suggesting that $NO_y$ concentrations from the sources did not readily remain near their origins but instead dispersed elsewhere, as reflected in the increasing transboundary contributions. However, this does not fully explain the concurrent increase in transboundary contributions, suggesting that other factors, such as the weakening of meteorological barriers, facilitated broader dispersion of $NO_y$ from source regions. These dynamics are discussed further below.

**%Source contributions to total NO$_y$ concentrations across East Asia**

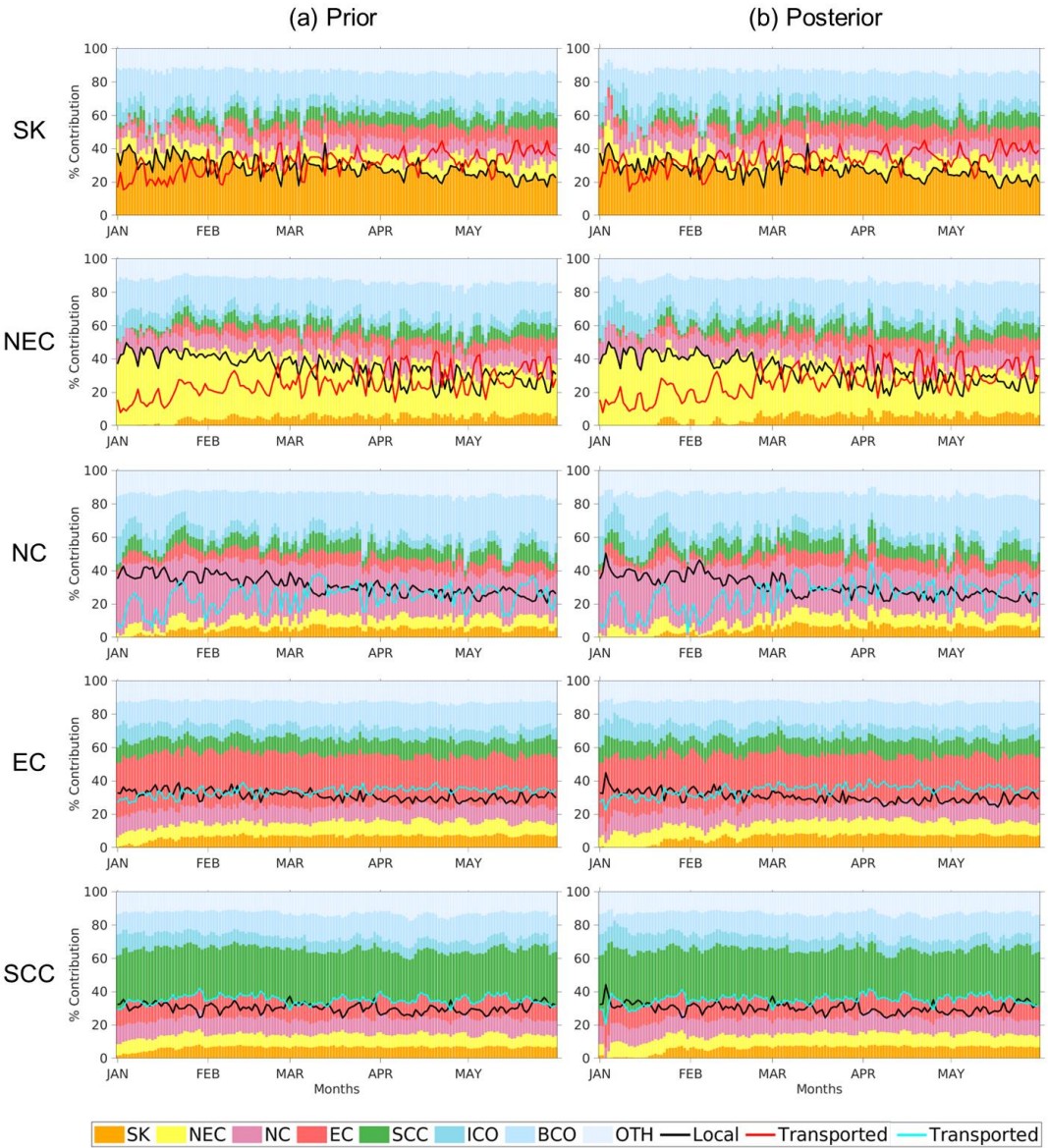

**Figure 4:** Percent contributions (%) of local and transboundary NO$_x$ emissions to NO$_y$ concentrations within the PBL in five source apportionment regions across East Asia during the period from January to May 2022. Prior and Posterior: percent contributions quantified based on the simulations using the a priori and a posteriori NO$_x$ emissions, respectively. ICO and BCO indicate the contributions from initial conditions and lateral boundary conditions, respectively, and OTH indicates the contribution of the emissions from the regions unspecified during ISAM.

**Table 2:** Descriptive statistics comparing the percent contributions (%) of local and transboundary $NO_x$ emissions to $NO_y$ concentrations within the PBL in five source apportionment regions across East Asia during the period from January to May 2022. Prior and Posterior: percent contributions quantified based on the simulations using the a priori and a posteriori $NO_x$ emissions, respectively.

| Receptor | Source | Model | January | February | March | April | May | Average |
|---|---|---|---|---|---|---|---|---|
| Korea | Korea (Local) | Prior | 34.54 | 29.54 | 29.33 | 26.23 | 23.76 | 28.68 |
| | | Posterior | 31.50 | 27.36 | 28.71 | 25.46 | 23.24 | 27.25 |
| | Northeast China | Prior | 8.34 | 8.84 | 9.18 | 9.14 | 8.96 | 8.89 |
| | | Posterior | 9.77 | 9.28 | 9.49 | 9.48 | 9.23 | 9.45 |
| | North China | Prior | 8.27 | 9.03 | 8.62 | 9.03 | 9.89 | 8.97 |
| | | Posterior | 9.63 | 9.93 | 9.11 | 9.41 | 10.25 | 9.67 |
| | East China | Prior | 3.52 | 6.36 | 8.02 | 8.89 | 9.79 | 7.32 |
| | | Posterior | 4.48 | 5.57 | 8.36 | 9.10 | 9.88 | 7.48 |
| | South Central China | Prior | 3.20 | 5.94 | 6.82 | 7.44 | 7.32 | 6.15 |
| | | Posterior | 3.28 | 6.85 | 7.34 | 7.89 | 7.53 | 6.58 |
| | (Transboundary total) | Prior | 23.33 | 30.17 | 32.64 | 34.50 | 35.97 | 31.32 |
| | | Posterior | 27.16 | 31.63 | 34.31 | 35.88 | 36.89 | 33.17 |
| Northeast China | Northeast China (Local) | Prior | 43.21 | 39.70 | 34.99 | 28.26 | 27.49 | 34.73 |
| | | Posterior | 42.62 | 38.93 | 32.97 | 26.86 | 26.29 | 33.53 |
| | Korea | Prior | 1.51 | 4.95 | 5.66 | 5.63 | 6.23 | 4.80 |
| | | Posterior | 1.07 | 2.72 | 6.39 | 6.43 | 6.73 | 4.67 |
| | North China | Prior | 9.75 | 8.46 | 8.58 | 9.62 | 9.80 | 9.24 |
| | | Posterior | 10.61 | 9.53 | 9.23 | 10.33 | 10.30 | 10.00 |
| | East China | Prior | 3.08 | 5.30 | 6.31 | 7.51 | 7.89 | 6.02 |
| | | Posterior | 2.45 | 4.00 | 6.93 | 8.00 | 8.23 | 5.92 |
| | South Central China | Prior | 3.10 | 5.32 | 5.79 | 6.29 | 6.76 | 5.45 |
| | | Posterior | 2.04 | 6.01 | 6.56 | 6.98 | 7.20 | 5.76 |
| | (Transboundary total) | Prior | 17.43 | 24.04 | 26.35 | 29.05 | 30.68 | 25.51 |
| | | Posterior | 16.17 | 22.25 | 29.11 | 31.73 | 32.46 | 26.35 |
| North China | North China (Local) | Prior | 37.36 | 35.34 | 30.36 | 27.30 | 25.83 | 31.24 |
| | | Posterior | 37.42 | 35.38 | 29.00 | 26.12 | 25.28 | 30.64 |
| | Korea | Prior | 2.68 | 5.16 | 5.94 | 5.60 | 5.58 | 4.99 |
| | | Posterior | 0.86 | 3.71 | 6.78 | 6.60 | 6.11 | 4.81 |
| | Northeast China | Prior | 5.12 | 5.44 | 6.78 | 6.60 | 6.33 | 6.06 |
| | | Posterior | 4.96 | 4.39 | 7.41 | 7.40 | 6.80 | 6.19 |
| | East China | Prior | 5.45 | 5.92 | 7.09 | 6.69 | 6.56 | 6.34 |
| | | Posterior | 5.41 | 5.03 | 7.69 | 7.45 | 6.96 | 6.51 |
| | South Central China | Prior | 5.69 | 6.32 | 7.39 | 7.16 | 7.40 | 6.79 |
| | | Posterior | 5.55 | 7.57 | 7.99 | 7.84 | 7.77 | 7.34 |
| | (Transboundary total) | Prior | 18.94 | 22.84 | 27.21 | 26.05 | 25.87 | 24.18 |
| | | Posterior | 16.79 | 20.69 | 29.87 | 29.29 | 27.64 | 24.86 |
| East China | East China (Local) | Prior | 33.66 | 32.74 | 31.15 | 28.91 | 29.71 | 31.24 |
| | | Posterior | 33.69 | 32.07 | 30.06 | 27.97 | 29.05 | 30.57 |
| | Korea | Prior | 4.21 | 7.09 | 7.20 | 7.23 | 7.27 | 6.60 |
| | | Posterior | 1.83 | 5.82 | 7.76 | 7.68 | 7.48 | 6.11 |
| | Northeast China | Prior | 7.92 | 7.62 | 8.36 | 8.87 | 7.86 | 8.13 |
| | | Posterior | 8.32 | 7.48 | 8.79 | 9.17 | 8.06 | 8.36 |
| | North China | Prior | 10.34 | 10.22 | 9.26 | 9.49 | 10.20 | 9.90 |
| | | Posterior | 10.76 | 10.62 | 9.64 | 9.85 | 10.55 | 10.29 |
| | South Central China | Prior | 8.69 | 8.59 | 9.36 | 9.77 | 9.35 | 9.15 |
| | | Posterior | 9.55 | 9.23 | 9.69 | 9.98 | 9.48 | 9.59 |

| | | | | | | | | |
|---|---|---|---|---|---|---|---|---|
| | (Transboundary total) | Prior | 31.16 | 33.52 | 34.17 | 35.35 | 34.68 | 33.78 |
| | | Posterior | 30.46 | 33.15 | 35.88 | 36.68 | 35.57 | 34.35 |
| South Central China | Southeast China (Local) | Prior | 31.22 | 30.52 | 30.89 | 28.23 | 30.40 | 30.25 |
| | | Posterior | 31.79 | 30.53 | 30.48 | 27.83 | 30.06 | 30.14 |
| | Korea | Prior | 5.10 | 7.06 | 6.92 | 6.79 | 6.70 | 6.51 |
| | | Posterior | 2.36 | 6.21 | 7.16 | 7.23 | 6.92 | 5.98 |
| | Northeast China | Prior | 7.87 | 7.31 | 7.57 | 7.87 | 7.19 | 7.56 |
| | | Posterior | 7.99 | 7.19 | 7.76 | 8.19 | 7.38 | 7.70 |
| | North China | Prior | 10.16 | 10.08 | 8.87 | 9.39 | 9.76 | 9.65 |
| | | Posterior | 10.54 | 10.32 | 9.08 | 9.77 | 10.05 | 9.95 |
| | East China | Prior | 11.84 | 12.30 | 10.65 | 11.10 | 11.43 | 11.46 |
| | | Posterior | 12.49 | 12.15 | 10.71 | 11.16 | 11.34 | 11.57 |
| | (Transboundary total) | Prior | 34.97 | 36.74 | 34.01 | 35.15 | 35.09 | 35.19 |
| | | Posterior | 33.39 | 35.87 | 34.71 | 36.34 | 35.70 | 35.20 |

The direction, reach, and amount of $NO_y$ transport varied noticeably from January to May, driven by seasonal synoptic settings that influenced prevailing winds. During the winter months, $NO_y$ transport was predominantly directed southeastward and eastward, seemingly consistent with the influence of the Siberian High, as shown by the presence of continental anticyclonic winds (Figure 5). This explains the relatively small transboundary contributions in upwind Northeast China and North China during this period (Table 2). Such transport pattern was more pronounced when using the a posteriori $NO_x$ emissions, showing an overall increase in transboundary $NO_y$ transport (Figure 5). After inversion, there was a noticeable increase in the amount of $NO_y$ traveling from the source regions towards the southeast and east, reaffirming the typical wintertime pollution transport patterns. During the spring months (March, April, and May) (Figure 6), the reach of $NO_y$ transport expanded westward noticeably, allowing pollutants to travel more freely across a broader range of directions, rather than being confined to the dominant southeastward flow seen in winter. This can be attributed to the weakening of the Siberian High and the associated northwesterly winds, which enabled a more dynamic and multidirectional transport of pollutants across the regions. In addition, the emergence of seasonal easterlies during this time can facilitate occasional westward transport of pollutants, especially when a low-pressure system travels from west to east across the region, passing to the south of the weakening Siberian High (Peterson et al., 2019). This pattern was particularly evident when using the a posteriori $NO_x$ emissions, as a broader reach of $NO_y$ transport was shown in all directions, with greater pollutant dispersal across the regions.

In addition, when using the a priori and a posteriori $NO_x$ emissions during spring, the severity of $NO_y$ pollution did not necessarily decrease as the distance from each source region increased, shown by some $NO_y$ hotspots far outside the sources (Figure 6). For example, when the SMA was the source region, we noticed some high $NO_y$ concentrations in highly populated regions in China such as Beijing, the Yangtze River Delta and Guangdong regions of China. Considering the "cooking time" allowed for $NO_x$ to become $NO_y$ components, which involves chemical reactions and transformations from $NO_x$ to longer-lived species such as $HNO_3$ and PAN (Shimadera et al., 2014; Yuan et al., 2018; Sun et al., 2020; Kashfi Yeganeh, 2024), we concluded that the high $NO_y$ loadings, particularly along the transport pathways, are secondary hotspots

(Figures S2 and S3); the formations of such species depend on the availability of radicals and other precursors that are often more abundant in urban environments, which emerged as these hotspots.

This expansion of transboundary contributions during the winter-spring transition reaffirms the critical role of seasonal dynamics in governing pollutant transport and dispersion, leading to increasingly complex cross-regional interactions. While some regions maintained consistent extents of transboundary contributions, others exhibited substantial fluctuations as the months progressed. For example, despite significant $NO_x$ emissions (Figure S2), Korea was initially the least influential source region, contributing approximately from 1% to 5% to transboundary $NO_y$ concentrations in neighboring regions during January due to the location relative to the dominant northwesterly. However, as the season transitioned, Korea's contribution to other regions grew, reaching up to 6%–7% by May, a notable increase comparable to the transboundary contributions shown by other regions (Table 2). Meanwhile, North China consistently emerged as the most influential source region, contributing around 10% to the $NO_y$ concentrations of neighboring regions throughout the months. Besides North China's dominance, East China and South Central China emerged as significant contributors to each other's $NO_y$ budget, reflecting a close transboundary relationship between these regions. In addition, South Central China consistently experienced substantial transboundary contributions that outweighed local contributions by around 5% throughout the entire study period, indicating the region's vulnerability to pollution transport. Even though the winter months offer meteorological conditions more favorable for the directional transport of pollutants, transboundary contributions to $NO_y$ loadings across the regions in this season were generally smaller compared to spring. This can be attributed to the stronger winds typical of winter, which can facilitate rapid transport of pollutants, but also can drive them to pass through receptor regions more quickly, limiting their accumulation (Wang et al., 2023). In contrast, during the spring months, the weakening of northwesterly winds seemed to allow for broader pollutant transport in a more multidirectional manner, leading to increased transboundary mixing of pollutants.

**Source contributions to total NO$_y$ concentrations across Asia: Winter 2022**

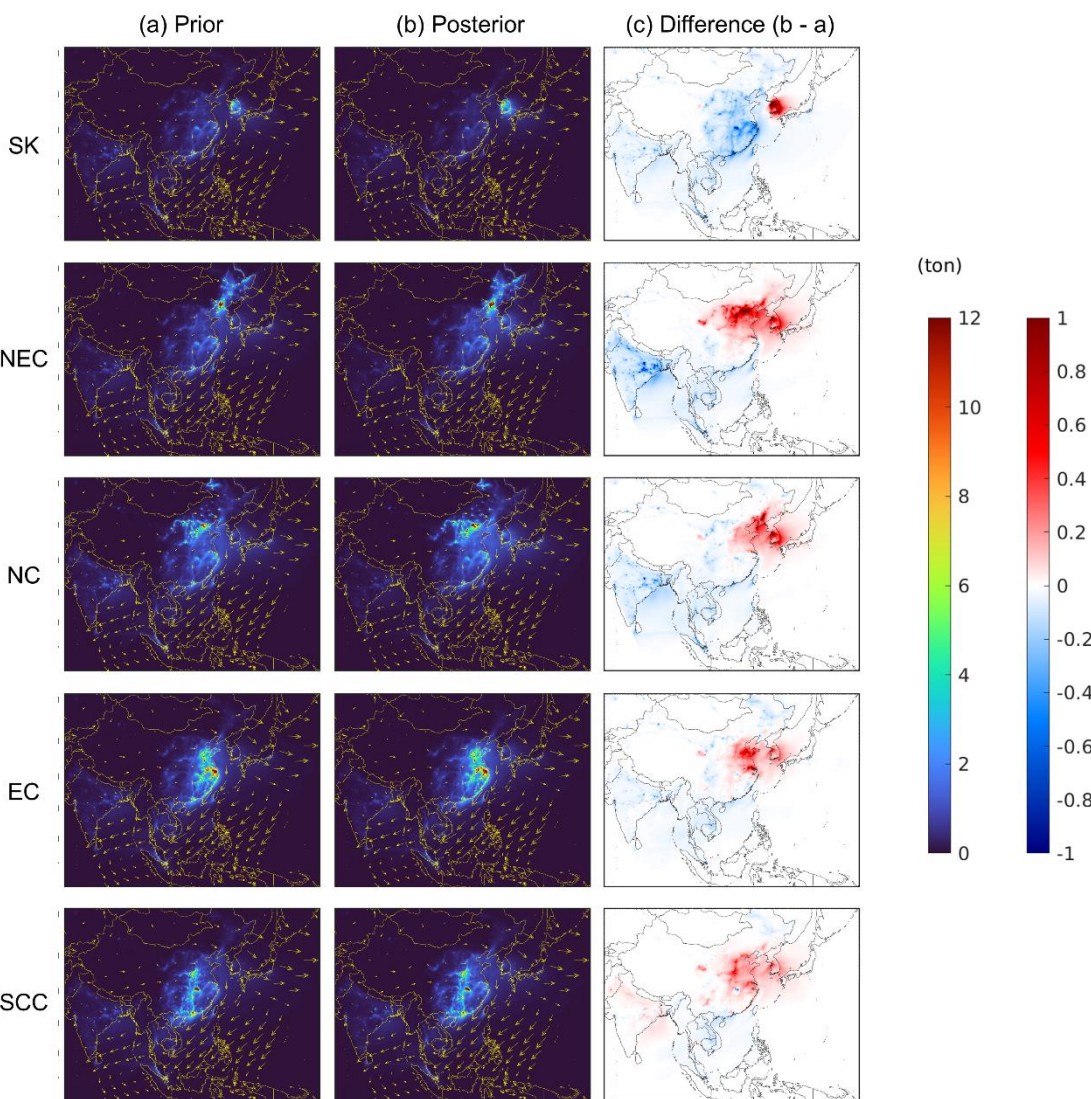

**Figure 5:** Source contributions to wintertime NO$_y$ concentrations (ton) within the PBL across Asia accumulated during the period from January to February 2022. (a) contributions quantified based on the simulations using the a priori NO$_x$ emissions, (b) contributions quantified based on the simulations using the a posteriori NO$_x$ emissions, (c) differences (b - a).

**Source contributions to total NO$_y$ concentrations across Asia: Spring 2022**

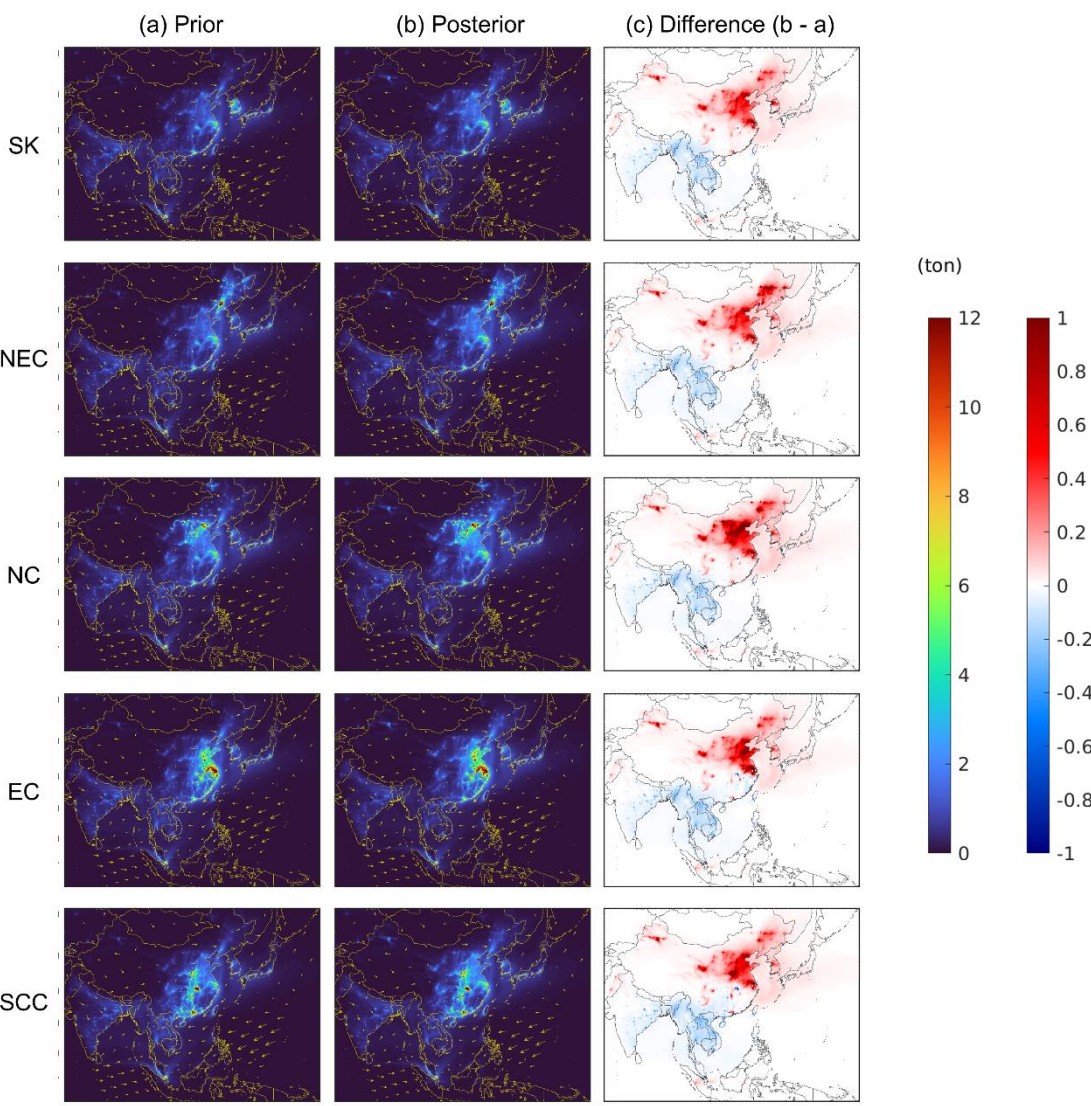

**Figure 6:** Source contributions to springtime NO$_y$ concentrations (ton) within the PBL across Asia accumulated during the period from March to May 2022. (a) contributions quantified based on the simulations using the a priori NO$_x$ emissions, (b) contributions quantified based on the simulations using the a posteriori NO$_x$ emissions, (c) differences (b - a).

### 3.3 Implications for PM$_{2.5}$ concentrations

While our study primarily focused on NO$_y$ loadings, the updates to the inventoried extent of NO$_x$ emissions also affected
surface PM$_{2.5}$ concentrations across East Asia. While the posterior model generally underestimated PM$_{2.5}$ concentrations in
Korea and China, the overall increases in NO$_x$ emissions after the inversion led to corresponding increases in PM$_{2.5}$
concentrations, which improved model accuracy to a certain extent in the regions (Figure 7; Table S3). In Korea, the use of
the a posteriori NO$_x$ emissions reduced the extent of model underestimation in PM$_{2.5}$ concentrations from -21.01% to -16.95%
on average during the winter-spring transition, leading to a slight improvement in IOA from 0.86 to 0.88, indicating better
alignment between modeled and observed concentrations in the region. The improvement was less pronounced in China,
where the model underestimation was only slightly reduced from -32.50% to -31.05%, with minor increases in R from 0.61
to 0.64 and IOA from 0.60 to 0.62. In both Korea and China, the increases in PM$_{2.5}$ concentrations seemed to be responsive
to the concurrent increases in nitrate aerosol concentrations, a major component of PM$_{2.5}$. For instance, our previous study
across East Asia (Park et al., 2023) suggested that secondary inorganic aerosols, such as nitrate, sulfate, and ammonium,
contributed around 53% of total PM$_{2.5}$ loadings in Korea, with nitrate aerosols alone accounting for 21%, on average during
the year 2019, while primary particulate matter made up around 47%. The improvements in our model accuracy suggest that
the overall upward adjustments to NO$_x$ emissions, which likely promoted nitrate aerosol formation, helped remedy the
model's prior underestimation of PM$_{2.5}$ concentrations. However, the underestimation still persisted after the inversion,
particularly in China, suggesting a possible underrepresentation of other PM$_{2.5}$ precursor emissions beyond NO$_x$, such as
sulfur dioxide (SO$_2$) and ammonia (NH$_3$), which were outside the scope of our study. Nonetheless, the response of PM$_{2.5}$
concentrations to the adjustments in NO$_x$ emissions reaffirmed the substantial contribution of secondary aerosols, such as
nitrate, to regional PM$_{2.5}$ concentrations across East Asia.

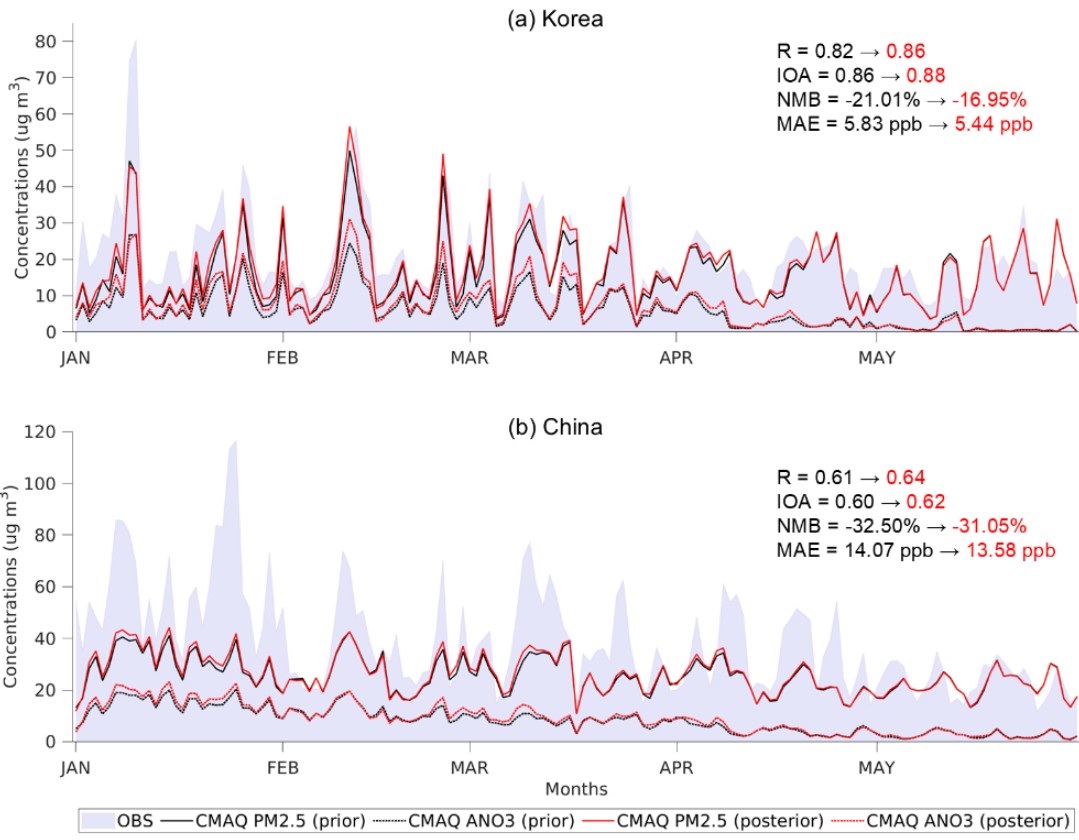

**Figure 7:** Daytime mean surface PM$_{2.5}$ concentrations observed and modeled at ground-based measurement sites during the period from January 1 to May 31, 2022. OBS: observed concentrations, CMAQ PM$_{2.5}$ (prior and posterior): modeled PM$_{2.5}$ concentrations using the a priori and a posteriori NO$_x$ emissions, CMAQ ANO$_3$ (prior and posterior): modeled nitrate aerosol (within PM$_{2.5}$ diameter) concentrations using the a priori and posteriori NO$_x$ emissions. Arrows indicate the changes in metrics from the prior model to the posterior model. R: Pearson's correlation coefficient, IOA: Index of Agreement, NMB: normalized mean bias (%), MAE: mean absolute error (ppb).

**4 Conclusions**

Our source apportionment effort, based on refined simulation accuracy, provided seasonal snapshots of $NO_y$ transport across East Asia during the 2022 winter-spring transition. Our diurnal updates to the $NO_x$ emissions inventory led to overall increases in $NO_x$ emissions in Korea and China by 50% and 33%, respectively. This suggests that the a priori estimates of $NO_x$ emissions from the global dataset might have underrepresented emissions patterns to a certain extent, demanding follow-up efforts to better account for local emissions in a more nuanced manner. These increases in the inventoried amount of $NO_x$ emissions substantially reduced the extent of the prior model's underestimation of surface $NO_2$ concentrations from -32.75% to -13.01% in Korea and from -10.26% to -3.04% in China, underscoring the utility of GEMS data as top-down constraints.

Leveraging the refined simulations, we quantified the local and transboundary contributions of $NO_x$ emissions to East Asia's $NO_y$ loadings during the winter-spring transition, focusing on our source apportionment regions including North China, Northeast China, East China, South Central China, and Korea. By comparing how much each region's $NO_x$ emissions contributed to its own $NO_y$ budget versus neighboring regions under seasonally varying synoptic settings, we assessed the cross-regional pollution transport dynamics and gained insights into source-receptor relationships across major $NO_x$-emitting regions of East Asia. During the winter months, pollutant transport was primarily influenced by strong northwesterly winds driven by the Siberian High, leading to significant transboundary contributions from upwind to downwind areas. As the Siberian High weakened in spring, transport patterns became more multidirectional, allowing pollutants to disperse farther across the regions. This seasonal transition resulted in increased transboundary contributions by up to 16% as the months progressed, as pollutants spread more extensively across the regions and potentially remained for an extended time near the receptors. From January to May, local contributions steadily decreased from 32%-43% to 23%-30%, while transboundary contributions showed an increasing trend from 16%-33% to 27%-37%. Some regions maintained their consistent contributions to East Asia's $NO_y$ loadings, whereas others showed noticeable fluctuations in the contributions as the months progressed. North China consistently contributed over 10% to other regions' $NO_y$ concentrations throughout the seasons, while Korea's contribution gradually increased from 1%-4% to 6%-7%, highlighting the critical role of seasonal synoptic conditions in governing pollution transport. This shift illustrates how a once less influential source region can become a significant contributor as seasons progress. While East China and South Central China substantially contributed to each other's $NO_y$ budget by 9%-12%, South Central China consistently experienced transboundary impacts that consistently exceeded its local contribution by 5%, indicating its vulnerability to pollution transport. These findings highlight the complexity of seasonal pollution dynamics and the evolving nature of transboundary impacts, underscoring the need for adaptive air quality management strategies that account for shifting transport patterns between emission sources and receptor regions.

A limitation of our study is that the source apportionment was largely confined to East Asia, despite our simulation domain covering the entirety of Asia. This limitation was primarily due to our study's objective, which necessitated improving model accuracy through indirect evaluation of the a posteriori emissions inventory's reliability, comparing modeled surface $NO_2$

concentrations against station measurements, which were available only for Korea and China during the study period. The absence of ground-based measurements from other regions in Asia during this period restricted the validation of the model's performance outside East Asia, limiting the broader applicability of our findings. Nonetheless, our study provides a comprehensive perspective on $NO_y$ transport dynamics, addressing the broader geographic context and extended simulation period that demand a more rigorous investigation. Future studies could build upon this approach by extending the study

period to cover additional seasons, such as summer and fall, to capture year-round pollution transport dynamics. A long-term, decadal study could reveal evolving trends in transboundary pollution, particularly during the winter-spring transition, and provide deeper insights into how policy changes, economic developments, and climate variations shape pollution patterns over time. These follow-up investigations would offer a more comprehensive understanding of East Asia's air quality challenges and further support the development of adaptive, long-term strategies for managing transboundary air pollution.

**Acknowledgments**

This work was partially supported by a grant from the National Institute of Environment Research (NIER), funded by the Ministry of Environment (MOE) of the Republic of Korea (NIER-2023-04-02-082). We thank the Research Computing Data Core at University of Houston for providing the supercomputing resources that supported this work.

**Data availability**

GEMS Level 2 products are available through Open-API at https://nesc.nier.go.kr/ko/html/svc/openapi/explain.do (in Korean) (last accessed on October 16, 2024), managed by the NIER's Environmental Satellite Center. Quality-assured AirKorea measurement datasets are available on the AirKorea website at https://www.airkorea.or.kr/web/last_amb_hour_data?pMENU_NO=123 (in Korean) (last accessed on October 16, 2024). MEE measurement datasets are available on the online archive at https://quotsoft.net/air/ (in Chinese) (last accessed on April

11, 2024), originally sourced from the China National Environmental Monitoring Center (CNEMC) database.

**Author contribution**

JP took the lead in drafting the original manuscript. JP and YC set up the experimental design. JP set up the models and conducted air quality simulations. JP and SK performed top-down adjustments to the emissions inventory. JP conducted source apportionment. JP, YC, and SK evaluated the emissions adjustment and source apportionment outcomes. YC

provided overall context as a principal investigator and supervised the entire research. All authors discussed the results and exchanged feedback to prepare the final version of the manuscript draft.

**Competing interests**

The authors declare that they have no conflict of interest.

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
