# Peer review of "Local and transboundary contributions to NOy loadings across East Asia using CMAQ-ISAM and GEMS-informed emissions inventory during the winter-spring transition"

_EGUsphere, 2024_

## Author Response (AR1)

**Referee #1**

The manuscript developed an updated NO$_x$ emission inventory in East Asia (constraint with GEMS) and analyze source-receptor relationship between China and S. Korea during 2022 winter-spring transition. The manuscript is well-written. The logic is clear, method being sound, and results are convincing. I only find few minor limitations need to address as follows:

| | |
|---|---|
| Comment | 1) line 28, it is better to provide the lifetime of NO$_x$ in days, cause the transport across yellow sea can also be short.

2) line 32. it is better to add some reference for long-range transport studies in Korea domain. there are many published during KORUS-AQ campaign, and you can compare your results here. |
| Response | Thank you for pointing these out. We acknowledge the need to further elaborate on the NO$_x$ lifetime and to provide additional context. We have revised the manuscript with additional details and references. |
| Changes made | L28-38: NO$_x$ has a relatively short atmospheric lifetime, typically ranging from a few hours … to several days … contributing to both local and transboundary air quality challenges. |

| | |
|---|---|
| Comment | 3) line 72, need to cite paper for Korea-China transport studies for example:

Eck et al, 2020 "Influence of cloud, fog, and high relative humidity during pollution transport events in South Korea: Aerosol properties and PM$_{2.5}$ variability"
Jordan et al, 2020 "Investigation of factors controlling PM$_{2.5}$ variability across the South Korean Peninsula during KORUS-AQ"
Nault et al, 2018, "Secondary organic aerosol production from local emissions dominates the organic aerosol budget over Seoul, South Korea, during KORUS-AQ"
Lee et al, 2020 "Sensitivity of simulated PM$_{2.5}$ concentrations over northeast Asia to different secondary organic aerosol modules during the KORUS-AQ campaign"
Choi et al, 2019. "Impacts of local vs. trans-boundary emissions from different sectors on PM$_{2.5}$ exposure in South Korea during the KORUS-AQ campaign"
Tang et al, 2023 "WRF-Chem quantification of transport events and emissions sensitivity in Korea during KORUS-AQ"
Bae et al, 2020 "Long-range transport influence on key chemical components of PM$_{2.5}$ in the Seoul Metropolitan Area, South Korea, during the years 2012-2016" |
| Response | Thanks for your suggestion and for sharing relevant references. We have incorporated these citations into our manuscript to enhance the introduction accordingly. |
| Changes made | L71-83: "Bae et al., (2020) investigated the influence of NO$_x$ … Similarly, Lee et al.'s study (2020) demonstrated that the long-range transport … Tang et al. (2023) also assessed the contributions of … 51% and 70% of the total PM$_{2.5}$ mass, respectively."

L127-135: "Beyond these satellite-based studies … Choi et al. (2019)'s study utilized in-situ observation data … Despite these efforts," |

| Comment | 4) line 90 to 120. need to mention Dr. Woo's KORUS-AQ v5 emission and also recent emission used for ASIA-AQ. those are state-of-art emissions widely used in research community and require an acknowledgement.

8) line 162, why do not use Dr. Woo's emission prepared for ASIA-AQ? it should be more accurate and widely used. if not, please address the reason in the manuscript. |
|---|---|
| Response | Thank you for the suggestion. We are aware of the exceptional utility and maturity of Dr. Woo's group's emissions inventory, having extensively utilized it in our previous studies. We chose to elaborate on this in the methodology section rather than in the introduction, where the focus is on the broader limitations of existing regarding the inventories (we do not intend to challenge the KORUS-AQ inventory with our 'top-down' estimates). Nevertheless, we have acknowledged the KORUS-AQ inventory and clarified our rationale for selecting the EDGAR inventory for our study. |
| Changes made | L211-219: "It is noteworthy that, unlike our previous study (Park et al., 2023) … was delicately built upon a combination of multiple regional surveys, providing comprehensive representations of emissions patterns across Asia … to avoid introducing additional uncertainties associated with the temporal allocation of monthly emissions into hourly emissions." |

| Comment | 5) line 120, please provide evidence than long-range transport only occur within PBL or rephrase. |
|---|---|
| Response | Thanks for pointing this out and we agree that the original description could mislead readers, as transport does not occur exclusively within the PBL. We have rephrased the relevant line and included additional clarifications in the methodology section. |
| Changes made | L143-145: "Then, using CMAQ-ISAM, we quantified the local and transboundary contributions of $NO_x$ emissions to $NO_y$ concentrations  across five major $NO_x$ source regions of East Asia during the period from January to May 2022.

L191-195: "However, it is important to acknowledge that substantial pollutant transport also occurs in the free troposphere beyond the PBL, where stronger winds facilitate long-range movement of pollutants. Our study specifically focuses on the PBL to assess cross-regional pollutant behaviors, as this layer directly influences surface air quality, the modeled estimates of which can be evaluated with station measurements (detailed in Section 2.5), and human health associated." |

| Comment | 6) line 130, need a brief description of what schemes, initial and boundary conditions you used here. I saw you sum them up in supplemental materials, but it is useful to introduce them in the manuscript as well. |
|---|---|
| Response | Thank you for your suggestion. We agree that including those details would improve the manuscript. We have accordingly enriched the methodology section, summarizing key configurations. |
| Changes made | L156-162: "We used the Morrison two-moment scheme for microphysics … from the National Centers for Environmental Prediction (NCEP) FNL operational model global tropospheric analysis."

 L175-180: "We used the YAMO scheme and the WRF omega formula … static boundary conditions during the entire simulation period." |

| Comment | 7) Figure 1. do you have plan to account N. Korea emission impact, it is much close to S. Korea and should have more close impact on S. Korea air quality? if not, please rephrase and explain reason. |
|---|---|
| Response | Thank you for your insightful concern. We recognize North Korea's proximity to South Korea and acknowledge the potential for its emissions to influence air quality in the region. Accurately accounting for North Korea's emissions poses is challenging due to the lack of reliable input data. Global emissions inventories like EDGAR, which rely heavily on accurate energy usage data to estimate pollutant emissions, have considerable uncertainty in regions where such data is unavailable or incomplete. This rationale has been clarified in the manuscript. |
| Changes made | L184-185: "North Korea, despite its close proximity to these regions and the potential impact of its emissions on neighboring regions' air quality … the lack of reliable input data for North Korea made it impractical to include as a separate source region in this study." |

| Comment | 4) line 90 to 120. need to mention Dr. Woo's KORUS-AQ v5 emission and also recent emission used for ASIA-AQ. those are state-of-art emissions widely used in research community and require an acknowledgement.

8) line 162, why do not use Dr. Woo's emission prepared for ASIA-AQ? it should be more accurate and widely used. if not, please address the reason in the manuscript. |
|---|---|
| Response | Thank you for the suggestion. We are aware of the exceptional utility and maturity of Dr. Woo's group's emissions inventory, having extensively utilized it in our previous studies. We chose to elaborate on this aspect in the methodology section rather than in the introduction, where the focus is on the broader limitations of existing inventories (we do not aim to challenge the KORUS-AQ inventory with our top-down estimates). We have acknowledged the contributions of the KORUS-AQ inventory and explained our rationale for selecting the EDGAR inventory for this study. |
| Changes made | L211-219: "It is noteworthy that, unlike our previous study … was delicately built upon a combination of multiple regional surveys … to avoid introducing additional uncertainties associated with the temporal allocation of monthly emissions into hourly emissions." |

| Comment | 9) line 208, "assume a linear relationship between $NO_2$ column and emission". linear relationship is not real. please provide your reason why assuming linear? or any reference to prove the method. |
|---|---|
| Response | Thank you for raising this point. We agree that further clarification is necessary regarding the assumption we made. The assumption of a linear relationship between $NO_x$ emissions and $NO_2$ concentrations is a widely adopted simplification in inverse modeling studies (listed in the manuscript). This approach is based on the understanding that satellite-observed $NO_2$ columns are primarily influenced by recent, localized $NO_x$ emissions due to the short lifetime of $NO_2$. Under these conditions, changes in $NO_x$ emissions are likely to result in proportional changes in $NO_2$ column densities, making the linearity assumption a practical method for constraining emissions. This assumption also simplifies the inversion process by directly linking observed $NO_2$ columns to emissions, avoiding the need for computationally intensive modeling of nonlinear chemical reactions and transport processes (in our study, the use of CMAQ DDM-3D partially accounts for the nonlinearity). We have clarified the rationale for this approach in the manuscript accordingly. |
| Changes made | L272-276: "This assumption, widely adopted in earlier inverse modeling studies… avoiding the need for computationally intensive modeling of nonlinear processes." |

| Comment | 10) line 212, nighttime $NO_2$ chemistry is very important for $NO_x$ distribution, giving that GEMS has no nighttime observation to constrain. please provide expected limitation of your study that ignore nighttime $NO_x$ chemistry. |
|---|---|
| Response | Thank you for the insightful suggestion. We agree that this limitation should be clarified more explicitly in the manuscript. We have added the relevant explanation in the manuscript accordingly. |
| Changes made | L303-305: "However, a limitation of this approach is that it does not improve the representation of nighttime emissions, leaving these unadjusted due to the absence of observational constraints. Consequently, any model biases associated with nighttime emissions remain unresolved." |

| Comment | 11) figure 2. when comparing GEMS $NO_2$ column with WRF-CMAQ model results, it is required that GEMS $NO_2$ is covered to use vertical profile from the same model. The default one for GEMS is NOT WRF-CMAQ. please provide how you do the conversion? |
|---|---|
| Response | Thanks for pointing this out. We have elaborated further on how we made the direct comparison between the GEMS's and CMAQ's $NO_2$ columns in the manuscript. |
| Changes made | L250-264: "including the averaging kernel, cloud fraction, data quality flags, … model-derived variables from the GEMS Level 2 data, including tropospheric and stratospheric air mass factors (AMF), a priori tropospheric $NO_2$ profile, and tropospheric pressure profile from the WRF model coupled with Chemistry (WRF-Chem) 3.9.1 (NIER, 2020). … excluded the cloudy scenes which could lead to inaccurate AMF references". |

**Referee #2**

In this manuscript, the authors report on a study investigating the contribution of transboundary transport to $NO_y$ levels in the Republic of Korea and China for January to May 2021. Using the CMAQ-ISAM, they first update the $NO_x$ emission inventory used by inverting GEMS $NO_2$ observations. They then separate the region of interest into 4 areas and use tagged $NO_y$ in the model to evaluate the contribution of different source areas to $NO_y$ levels in East Asia during the winter spring transition.

The manuscript discusses an interesting topic. It is well structured and clearly written, and the methods and conclusions appear valid. However, there are several critical aspects which need to be addressed and clarified before the manuscript can be accepted for publication.

**Major comments:**

| Comment | In the title and in other parts of the manuscript, the authors talk about "nitrogen loading". When I first read this, I expected that this would include ammonia and particulate nitrates, but this is not the case. Please use more precise wording to make clear what exactly is covered by this study. |
|---|---|
| Response | We agree that our wording "nitrogen loading" may mislead readers into expecting a broader scope, including species like ammonia and nitrate aerosols, which are not part of this study. We have revised the title and manuscript for clarity, accordingly. |
| Changes made | Title: Local and transboundary contributions to $NO_y$ loadings across East Asia … during the winter-spring transition
L8: "… to reactive nitrogen species ($NO_y$) loadings across East Asia …"
L140: "… East Asia's $NO_y$ concentrations during the winter-spring …"
L479: "… primarily focused on $NO_y$ loadings …"
L507: "… provided seasonal snapshots of $NO_y$ transport across East Asia …"
L543: "… a comprehensive perspective on $NO_y$ transport dynamics, |

| Comment | If I understood the manuscript right, the definition of $NO_y$ used does not include $NO_x$. This makes sense if one would like to focus on transport, but it can be misleading if the results are interpreted in the way of "how much of the reactive nitrogen pollution is due to transnational transport". I think it would be better to include $NO_x$ in $NO_y$. If that's not possible, the authors should at least include a column in the tables giving the fraction that transported $NO_y$ contributes to $NO_y + NO_x$ assuming the latter is local. |
|---|---|
| Response | Thank you for pointing out this critical oversight. Our definition of $NO_y$ includes $NO_x$, but we mistakenly did not state this. We have corrected this in the manuscript accordingly. |
| Changes made | L196-197: "Note that we used the summation of $NO_x$, nitric acid ($HNO_3$), nitrous acid (HONO), and peroxyacetyl nitrate (PAN) to represent $NO_y$ …" |

| Comment | I'm completely confused by Figures 5 and 6. If I did not misunderstand the figures, they are supposed to show the absolute mixing ratio of $NO_y$ that is present throughout East Asia and originates in one of the selected source regions. However, the maps show clear hotspots in polluted regions such as SMA even when the contribution is from another source region. Shouldn't these maps show smooth distributions outside the source regions? And shouldn't we see a reduction with distance from the source region? Please explain or correct the figures. |
|---|---|
| Response | We acknowledge that the discussions accompanying Figures 5 and 6 do not fully address the rationale behind the spatial distributions of $NO_y$, particularly the occurrence of hotspots outside the source regions.

We'd like to clarify that these figures do not solely represent the physical transport of $NO_y$ across the domain but rather illustrate where each source region's $NO_x$ emissions contribute to local and transboundary $NO_y$ concentrations. The formation of $NO_y$ involves chemical reactions and transformations, which can lead to secondary hotspots outside the immediate vicinity of the source regions.

To better understand these hotspots, we examined the spatial distributions of individual $NO_y$ components. The earlier $NO_y$ hotspots along the transport pathways seemed to correspond to longer-lived species such as $HNO_3$ and PAN, the formations of which depend on the availability of radicals and other precursors that are often more abundant in urban environments, which led to those hotspots pointed out. We have revised the manuscript to provide a more detailed explanation to support our discussions, accordingly. |
| Changes made | L433-441: "In addition, when using the a priori and a posteriori $NO_x$ emissions during spring, the severity of $NO_y$ pollution did not necessarily decrease as the distance from each source region increased… the formations of such species depend on the availability of radicals and other precursors that are often more abundant in urban environments, which emerged as these hotspots." |

| Comment | I do not follow the discussion on the impact of wind speed on the fraction of transported $NO_y$. In the manuscript, it is claimed that low wind speeds lead to accumulation of pollutants (so far, I agree) and that this leads to a larger contribution of transported $NO_y$. This I do not understand, as accumulation (and dilution) will work in a similar way for locally produced and transported $NO_y$, and therefore, the ratio is not changed. In my opinion, the only relevant quantities are a) how much $NO_y$ is transported over the borders into the selected region and b) how much $NO_y$ is produced within this region. Please explain.

(From Minor comments) Lines 368 and 373: See my major comment on the impact of wind speed. I do not agree with the arguments made here about |
|---|---|

| | |
|---|---|
| | "rapid passing through" and "longer lingering" as wind speed affects all $NO_y$, not just the transported $NO_y$. |
| Response | Thank you for pointing this out. We acknowledge that the original discussion was self-contradictory in attributing a larger contribution of transported $NO_y$ to weaker seasonal winds. Particularly, our statements were overdoing the relationship between the weakening Siberian High and slower exit of $NO_y$ without supporting evidence. We recognize that accumulation due to weaker winds would affect both transported and locally produced $NO_y$ similarly, and thus, the ratio between the two would not necessarily change as you commented.

To address this, we have removed the unnecessary and misleading explanations in the discussion and revised the text to focus on the absolute quantities of $NO_y$ transported across borders and produced locally within the selected regions. This clarification aligns with the reviewer's valid observation and ensures that the discussion reflects a more accurate interpretation of the results. |
| Changes made | L425: "… expanded westward noticeably …"

L372-374 (in the original manuscript): ""

L376-378 (in the original manuscript): ""
" |

**Minor comments:**

| Comment | L149: While the PBL is an interesting region for air pollution studies, I'm surprised that the authors assume that it is also the altitude region where the relevant transport processes take place. |
|---|---|
| Response | Thanks for pointing this out and we agree that the original manuscript could mislead readers, as transport does not occur exclusively within the PBL. We have rephrased the relevant line and included additional clarifications in the methodology section. |
| Changes made | L191-195: "However, it is important to acknowledge that substantial pollutant transport also occurs in the free troposphere beyond the PBL, where stronger winds facilitate long-range movement of pollutants. Our study specifically focuses on the PBL to assess cross-regional pollutant behaviors, as this layer directly influences surface air quality, the modeled estimates of which can be evaluated with station measurements (detailed in Section 2.5), and human health associated." |

| Comment | L198: The sentence on the use of averaging kernels is rather vague and should be formulated more precisely. |
|---|---|
| Response | Thanks for pointing this out. We have elaborated further on how we made the direct comparison between the GEMS's and CMAQ's $NO_2$ columns in the manuscript. |
| Changes made | L248-264: "… including the averaging kernel, cloud fraction, data quality flags, … model-derived variables from the GEMS Level 2 data, including tropospheric and stratospheric air mass factors (AMF), a priori tropospheric NO2 profile, and tropospheric pressure profile from the WRF model coupled with Chemistry (WRF-Chem) 3.9.1 (NIER, 2020). … excluded the cloudy scenes which could lead to inaccurate AMF references". |

| Comment | L219: Note that GEMS does not take instantaneous snapshots of $NO_2$ but scans for 30 minutes from East to West. |
|---|---|
| Response | Thanks for your comment. We realized that our original description gives the nuance like GEMS's retrievals occur at such intervals. We have corrected the relevant statement in the manuscript accordingly. |
| Changes made | L285: "Due to the 15-minute offset in the availability of GEMS Level 2 products (from 22:45 to 07:45 UTC)," |

| Comment | L261: Introduce SMA |
|---|---|
| Response | We have added a description of the SMA earlier in the manuscript to provide some context before the discussion. |
| Changes made | L182-184: "During discussions for Korea (later in Section 3), we focused on the SMA, the country's economic hub, where dense traffic activity contribute to severe air pollution (Figure 1) (Park and Lee, 2020)." |

| Comment | Section 2.3: It is well known from validation studies that version 2 of the GEMS tropospheric $NO_2$ product as well as GEMS cloud fractions still have some issues. This should be briefly discussed and the impact on emissions and the quantification of $NO_y$ transport be mentioned. |
|---|---|
| Response | Thanks for pointing it out, and we have added further explanation to support our use of the retrieval data that have a cloud fraction < 0.3, the criteria of which (as well as the problematic aspect in the dataset you have noted) was suggested in an earlier study that used GEMS Level 2 data version 2.0 (Lange et al., 2024). |
| Changes made | L262-264: "…note that since the Level 2 data version 2.0 quit employing the OMI climatology thereby deserves further validation efforts through retrieval studies, we excluded the cloudy scenes which could lead to inaccurate AMF references." |

| Comment | Figure 2: Are the model fields sampled at the times of valid GEMS measurements? |
|---|---|
| Response | Yes, we did not use any modeled scene outside the GEMS's retrieval times mentioned in the Methods to ensure their nearly fair comparisons against the retrieval data. We have clarified this aspect in the caption of the Figure 2. |
| Changes made | Figure 3 caption: "… Note that we excluded the modeled columns that do not correspond with the GEMS's retrieval times." |

| Comment | Figure 2: I do not understand why the model with a posteriori emissions overestimates $NO_2$ over large regions of Northern and Central China and Korea in February and March. With the assumption of local linearity, I would have expected the model to always do a good job on local hotspots such as SMA. |
|---|---|
| Response | Thanks for raising this important aspect regarding the overly adjusted a posteriori values. The modeled columns theoretically must become closer to the observed columns, but our employment of the Bayesian approach, which |

| | is often regarded as a simple inverse modeling method, could not fully resolve the transport effect (non-locality in the real world). We have added explanation for this aspect in the manuscript, accordingly. |
| --- | --- |
| Changes made | L345-348: "In addition, we noticed some posterior overcompensation in the modeled columns, shown by some overestimated values across North and South Central China and Korea … a simple inverse modeling method which cannot fully resolve the non-locality of air pollutants (Park et al., 2024) …" |

| Comment | Figure 2: What does the term "hourly" in the figure caption refer to? |
| --- | --- |
| Response | Thank you for pointing it out, and we realized that the caption is a little bit confusing. "Monthly average of hourly tropospheric $NO_2$ columns" means the average of hourly tropospheric $NO_2$ columns for each month from January to May. We have revised the caption, accordingly. |
| Changes made | Figure 2 caption: "Averages of hourly tropospheric $NO_2$ columns (molecules/cm$^2$) observed and modeled during daylight hours (GEMS retrieval hours) in each month from January to May 2022". |

| Comment | Line 323: A decline in anthropogenic emissions will increase the relative contribution of transported $NO_y$ only if the other regions do not see a similar decline in emissions. |
| --- | --- |
| Response | Thank you for your comment. We realized that part of our earlier statement might have been misleading to readers. We have revised it accordingly in the manuscript. |
| Changes made | L403-405: "However, this does not fully explain the concurrent increase in transboundary contributions, suggesting that other factors, such as the weakening of meteorological barriers, facilitated broader dispersion of $NO_y$ from source regions. These dynamics are discussed further below." |

| Comment | Figure 4: Introduce ICO, BCO, OTH |
| --- | --- |
| Response | Thank you for pointing out the missing descriptions. We have introduced ICO, BCO, and OTH in the caption. |
| Changes made | Figure 4 caption: "ICO and BCO indicate the contributions from initial conditions and lateral boundary conditions, respectively, and OTH indicates the contribution of the emissions from the regions unspecified during ISAM". |

| Comment | Figure 5: The ppb scale does not seem correct to me – 5x10E4 ppb of $NO_y$? |
|---|---|
| Response | Thanks for pointing it out, and we realized that our use of the ppb scale does not make sense when it comes to representing the summation of the pollutant loadings. We have corrected the unit from ppb to ton, which better explains such an aspect, accordingly. |
| Changes made | Figures 5, 6, and S2 units |

| Comment | Figure 5: What does the term total (sum) $NO_y$ imply? |
|---|---|
| Response | The term indicates the total amount of $NO_y$ accumulated during the winter months from January to February. We acknowledge that the term may confuse readers, so we have revised the captions. |
| Changes made | Figures 5 and 6 captions |

---

## Author Response (AR2)

**Editor decision: Publish subject to minor revisions (review by editor)**

Dear authors,

Please find enclosed a further referee report on the revised version of your manuscript. Although the referee suggest publication of the manuscript as is, I still have some minor revision I would like to ask you to consider before the manuscript can also be accepted from my side.

| Comment | P1, L11: What do you mean with informed? Is that really the right term? Do you mean "assimilated"? |
|---|---|
| | P1, L12: Same here in "GEMS-informed". Further, why is it always "our" Baysian inversion? Isn't this just a "Bayesin inversion", thus shouldn't it rather read "the" or "a". |
| Response | The $NO_x$ emissions used for the simulations were the top-down estimates derived from the Bayesian inverse modeling using GEMS observation data. As you pointed out, the term 'informed' seems to not clearly deliver this nuance. We have revised the manuscript accordingly. |
| Changes made | L11: "…$NO_x$ emissions adjusted by the Geostationary…
L12: "After the Bayesian inversion…"
L122: "…top-down constraints during the Bayesian inversion…"
L153: "…the Bayesian inversion…"
L347: "……which the Bayesian inversion…" |

| Comment | P1, L13-15: The sentence with the emission increases is not clear. Do you refer here to results from an earlier study? To what compared did you derive an underestimation with the model? Please rephrase the sentence and provide a clear statement. |
|---|---|
| Response | Thanks for pointing it out and we agree that the sentence was not clear. We have rephrased the sentence accordingly. |
| Changes made | L14: "After the Bayesian inversion, the inventoried $NO_x$ emissions increased by 50% in Korea and 33% in China compared to the a priori estimates…" |

| Comment | P3, L95 and further occasions throughout the text: emissions inventory -> emission inventory |
|---|---|
| | P6, L186: emissions inventories -> emission inventories |
| Response | We have revised our manuscript accordingly for consistency. |

| Comment | P1-5: The introduction is with almost four pages to long. The topic if your study is not that complicated that it justifies such a long introduction. Please shorten to 2-3 pages. |
|---|---|
| Response | We acknowledge that our introduction was lengthy. We have made it more concise while retaining key literature, ensuring the length remains within three pages. |
| Changes made | L46-48: ""

L64-72: ""

L83-84: ""

L87-90: ""

L128-136: "" |

| Comment | P5, L149: Rename "Materials and Methods" to "Data and Methods"

P5, L150: You actually do not provide here any model description, rather the simulation set-up is described. Thus, this section should be renamed to "Model simulations"

P5, L153 and throughout the manuscript: "Figure" and "Section" are abbreviated in the text as "Fig." and "Sect." unless they appear at the begin of the sentence (see ACP manuscript preparation guidelines). |
|---|---|
| Response | Thank you for pointing these out. We have renamed the sections and adjusted abbreviations as suggested. |

| Comment | P6, L183: "During discussions for Korea....." please rephrase the sentence part. |
|---|---|
| Response | We have rephrased the sentence for better clarity. |
| Changes made | L 161-163: "In our discussion of Korea (later in Sect. 3), we focused on the SMA, the country's economic hub, where dense traffic activity contributes to severe air pollution, as highlighted in the small panel of Fig. 1, which illustrates its geographic extent." |

| Comment | P7, L193: movement -> transport
P7, L203: This section should be renamed to " Emission inventories".
P8, L237: CB6 -> CB06 (to be consistent with the writing of CB05) |
|---|---|
| Response | We have revised our manuscript as suggested accordingly. |

| Comment | P25, Figure 7 title: Adjust font size for "across".
P25, Figure 7 and Figure 7 caption: What is ANO3? I guess you mean "HNO3"? |
|---|---|
| Response | Thanks for pointing it out. Earlier we used CMAQ's output variable name ANO3 (aerosol nitrate) but now replaced it with $NO_3^-$ for better clarity. |

| Comment | Reference list: In the author lists "&" should be replaced by "and"

P30, L600: All author names should be listed. |
|---|---|
| Response | We have corrected the format in the reference list. |